# Current Challenges and Opportunities of Photodynamic Therapy against Cancer

**DOI:** 10.3390/pharmaceutics15020330

**Published:** 2023-01-18

**Authors:** Ruben V. Huis in ‘t Veld, Jeroen Heuts, Sen Ma, Luis J. Cruz, Ferry A. Ossendorp, Martine J. Jager

**Affiliations:** 1Department of Ophthalmology, Leiden University Medical Centre (LUMC), 2333 ZA Leiden, Zuid-Holland, The Netherlands; 2Department of Radiology, Leiden University Medical Centre (LUMC), 2333 ZA Leiden, Zuid-Holland, The Netherlands; 3Department of Immunology, Leiden University Medical Centre (LUMC), 2333 ZA Leiden, Zuid-Holland, The Netherlands

**Keywords:** photodynamic therapy, cancer, reactive oxygen species, immunogenic cell death, damage-associated molecular patterns, photoimmunotherapy

## Abstract

BACKGROUND: Photodynamic therapy (PDT) is an established, minimally invasive treatment for specific types of cancer. During PDT, reactive oxygen species (ROS) are generated that ultimately induce cell death and disruption of the tumor area. Moreover, PDT can result in damage to the tumor vasculature and induce the release and/or exposure of damage-associated molecular patterns (DAMPs) that may initiate an antitumor immune response. However, there are currently several challenges of PDT that limit its widespread application for certain indications in the clinic. METHODS: A literature study was conducted to comprehensively discuss these challenges and to identify opportunities for improvement. RESULTS: The most notable challenges of PDT and opportunities to improve them have been identified and discussed. CONCLUSIONS: The recent efforts to improve the current challenges of PDT are promising, most notably those that focus on enhancing immune responses initiated by the treatment. The application of these improvements has the potential to enhance the antitumor efficacy of PDT, thereby broadening its potential application in the clinic.

## 1. Introduction

After administration, the photosensitizer (PS) distributes to the tumor and is illuminated with light, often non-thermal light, in the red too-deep red light spectrum (Figure 1) [1]. The energy of the light is then absorbed, moving the PS from its energetic ground state to the excited singlet state. From there, it may undergo intersystem crossing to enter the excited triplet state and subsequently transfer its energy to yield ROS, most often singlet oxygen. This PDT-induced ROS production can subsequently initiate cell death in a manner that has been shown to attract immune cells of myeloid and lymphoid origin.

## 2. History of Photodynamic Therapy of Cancer

It is generally assumed that photodynamic therapy is a relatively novel therapeutic option. However, the major roots of photodynamic treatment date back to the early 20th century. The scientific basis of PDT was laid by Oscar Raab in the group of von Tappeiner [2,3], who discovered during a thunderstorm that the combination of an acridine red dye and light killed a single cellular species of Paramecium, a genus of eukaryotic unicellular ciliates. Progress for PDT toward cancer treatment was made in 1948 when Figge summarized several studies that reported selective accumulation of porphyrins in murine tumors [4]. Another leap was made when clinicians at the Mayo Clinic observed an enhanced fluorescence in tumors when using a derivative of hematoporphyrin [5,6], which is an acetic acid-sulfuric acid derivative of hematoporphyrin hydrochloride termed hematoporphyrin derivative (HPD), of which the specific content was unknown at the time. Hematoporphyrin is an iron-containing porphyrin that can be created from a heme group, e.g., hemoglobin in the blood. Porphyrins are a group of organic compounds with a tetrapyrrole structure that can easily absorb light due to their structural conformation. In the following years, HPD was used to localize neoplasia by fluorescence in various patients [7]. The first report describing the therapeutic effects of PDT was provided by Diamond et al. showing, in a rather basic setup, that glioma cells could be destroyed in culture or in animal models [8]. Similar results were obtained by Thomas Dougherty, who induced complete responses in tumor-bearing mice and rats using hematoporphyrin and a slightly more advanced light setup [9]. Expanding on these preclinical data, the group of Dougherty successfully treated patients with numerous tumor types in various locations, using HPD as their sensitizer of choice and designating the treatment as ‘photodynamic therapy’ [10,11]. The biodistribution was determined by radiolabeling HPD and monitoring its distribution in patients [12], showing localization to tumors in addition to the kidneys, liver, and spleen. As such deep-seated organs are protected from treatment light, they assigned PDT a measure of tumor-specificity. During the same period, the group of Dougherty also determined that singlet oxygen is the major cytotoxic agent that is generated during PDT [13]. They continued to demonstrate the antitumor efficacy of HPD, eventually resulting in the approval of the treatment by the Food and Drug Administration (FDA). At the time, no mechanism behind the cytotoxicity toward malignant cells was determined. However, the ability of PDT to induce vascular disruption and shutdown was recognized [14], leading to the use of verteporfin in the treatment of macular degeneration [15]. Together with vascular endothelial growth factor (VEGF) antagonists, PDT remains a treatment for this illness. After this, one study identified that only dimers and higher oligomers were the active components present in HPD [16]. This finding led to the synthesis of the FDA-approved and commercially available Photofrin, consisting of HPD without monomeric porphyrins. Since then, many different sensitizers and PDT protocols have been compared, attempting to identify the optimal PS for PDT in the clinic.

Another major advance was the discovery of apoptosis as a major mechanism behind PDT-induced cytotoxicity [17]. Subsequent studies found that apoptotic factor Bcl-2 was a common target for photodynamic damage [18,19] and that necrosis and autophagy can also occur after PDT. Following this, other types of PDT-induced cell death have been identified, the occurrence of which depends on the sensitizer and protocol used [20]. It was then discovered that cell death could be accompanied by the release and exposure of molecules with inflammatory properties. Such inflammatory molecules were designated damage-associated molecular patterns (DAMPs) [21,22,23,24] which were also found to be released after PDT [22,25,26,27,28,29,30,31]. Moreover, it was shown that PDT could lead to a type of cell death that can induce adaptive immune responses, a process that was called immunogenic cell death (ICD) [32,33]. Many studies have reported that PDT has been shown to induce ICD through the release of damage-associated molecular patterns (DAMPs) [20]. These observations led to the discovery of a third mechanism behind PDT-induced antitumor efficacy: the induction of antitumor immune responses. Rather than focusing on optimizing tumor cell death and disruption of the vasculature, PDT-related research is currently shifting towards optimizing ICD and antitumor immune responses by combining PDT treatment with immunotherapy. The primary aim of PDT in this approach is to debulk the tumor and/or to damage its vasculature while initiating an acute inflammation in the tumor that can be further strengthened by immunotherapeutic agents, which has been shown to result in adaptive antitumor responses capable of inducing abscopal effects (tumor cell death in a lesion elsewhere in the body). A major advantage of this strategy is that tumors that cannot be reached by light sources can be treated with immunotherapy, while PDT enhances the magnitude of the immune response while reducing the size of tumors to facilitate the entry of immune cells and immunotherapeutic agents. However, not all combinations of PDT and immunotherapy are equally effective against different tumors, and successful treatment is highly dependent on the careful selection of immunotherapeutic agents, sensitizers, PDT protocols as well as the timing of the treatment.

## 3. The Fundamentals of Photodynamic Therapy: Sensitizers, Light Penetration in Tissues, Light Sources, Photodynamic Effect

### 3.1. Photosensitizers

In cancer treatment, photosensitizers (PS) are molecules that transfer the energy of light to change molecular structures proximal to their location at the time of illumination. When sufficient changes occur in a cancer cell, they can become unrepairable and may subsequently initiate cell death. Most clinically used PS is based on the tetra-pyrrole structure and are derivatives of porphyrins, chlorines, or dyes. Over the years, several generations of PS have been developed, of which the very first generation includes HPD from the time of Oscar Raab. The second generation of PS (e.g., hypericin, phthalocyanines, chlorins, benzoporphyrin derivatives, protoporphyrin IX, etc.) was designed to overcome the limitation of HPD, including improved absorption in the red to deep-red spectrum and enhanced biodistribution after administration. The third generation of PS involves multifunctionality of PS, including the utilization of passive of active targeting strategies for tumors or to subcellular locations, by association with or coupling to nanoparticles, antibodies, and ligands specific to tumor targets. The fourth generation of PS, although they are technically not always sensitizers, involves the use of porous carriers for sensitizers, such as metal-organic frameworks and mesoporous silica that can be loaded with a large number of PS. An overview of the characteristics of several notable and clinically used PS that have been under extensive investigation is given in Table 1. The different PS possess various chemical characteristics, which enable their use for different indications. For instance, some PS remains mostly in the vasculature after intravenous administration and are therefore optimal for indications that require a vascular shutdown or antitumor strategies that focus on disruption of the tumor vasculature. Other PS, however, can easily exit the vasculature and enter the tumor, where they are taken up by cancer cells and induce cytotoxicity after illumination. Additional differences between PS include the wavelength required for optimal induction of the photodynamic effect. Although most PS are selected to absorb light in the red-to-deep-red spectrum due to the attenuation of light in tissues (explained elsewhere in this Thesis), there are exceptions that absorb light at different wavelengths. In addition, there is an inverse correlation between the wavelength and energy of the light. Therefore, sensitizers that absorb light at different wavelengths will have access to light with different energy. Related to this, there are differences in the efficiency of energy transfer from light to oxygen radicals, regardless of the energy of the light. The ability of a PS to generate damage-inducing molecules is often expressed as their singlet oxygen quantum yield, which can be measured and compared. Furthermore, there are differences in the solubility of PS, and certain sensitizers require other molecules, e.g., nanocarriers, to enable their use in cancer treatment. Additional important differences can be found in the route of entry into cells, preferential subcellular locations of the PS that will subsequently undergo the effects of PDT-induced damage, toxicity in the absence of light, retention of PS in off-target areas, e.g., the skin and ability of the PS to induce immunogenic cell death with the ability to initiate antitumor immune responses. Although the ideal PS has not yet seen the light of day, the characteristics of such a sensitizer have been described [34,35,36]. Briefly, it displays no or little toxicity in the absence of light, referred to as dark toxicity. Illumination should induce localized damage throughout all areas of the tumor and possibly its vasculature. The ideal PS will induce strong antitumor immune responses with the ability to clear the remaining tumor cells, as well as distant metastatic lesions. A favorable biodistribution should facilitate little off-target effects, no or minimal skin photosensitivity, and rapid clearance from the body after the antitumor effect has been established.

### 3.2. Drug-to-Light Interval

Several other factors and concepts are important to consider for PDT treatment. For example, post-administration of a PS, illumination with a light source must occur to induce a photodynamic effect. This is performed at a certain time point after administration, which is known as the drug-to-light interval (DLI). The DLI for PDT is often based on the pharmacokinetics of the PS and chosen to induce damage to the areas of interest: the tumor, the vasculature, or a mixture of both. For PS that distribute to the tumor after intravenous administration, using a short DLI would target mostly the vasculature, whereas a long DLI would mostly target the tumor cells, and a medium DLI would target both.

### 3.3. Attenuation and Propagation of Light in Tissues

Another consideration for PDT treatment is the maximum tissue penetration depth of the light used for therapy. As noted, the wavelength of light sources required for PDT is determined by the absorption spectrum of the PS used for therapy. Most PS, e.g., chlorin e6, possess a large Soret-absorption band at shorter wavelengths in addition to one or several smaller Q-bands that usually fall in the (deep) red spectrum. The smaller Q-band is most often used for PDT, as it has a deeper tissue penetration depth due to the optical properties that attenuate light in tissues intended for treatment. Such attenuation of light through tissues can be attributed to reflectance, absorption, scattering, and refraction. The impact on loss of light density by reflection and refraction is dependent on the relative values of the refractive indices and are proportional to the angle of light between two media [37]. Therefore, their impact can be minimized by perpendicular illumination towards the interface of two different media. However, the scattering of light in tissues represents the strongest influence on the attenuation of light intensity as well as on light directionality [38], reducing the penetration depth of light in tissues such as tumors. Absorption of light also has a large impact on the reduction in light intensity [38], and the absorption spectra of the chromophores present in tissues, most notably water, oxy- and deoxyhemoglobin, melanin, and various cytochromes, determine the optimal spectrum in terms of light penetration [36,37,39].

### 3.4. Therapeutic Window of PDT

Light only penetrates tissues to a certain depth. The optimal optical region in tissues, as mentioned before, spans approximately 600–1200 nm and is known as the therapeutic window for light delivery that allows optimal light penetration through tissues. However, wavelengths higher than approximately 800 nm generally contain insufficient energy to generate a strong photodynamic effect [40] and require solutions such as upconversion of photons for sufficient singlet oxygen quantum yields. For these reasons, most clinically used PS are illuminated at wavelengths in the 600–800 nm range. However, even when using light within the therapeutic window, the tumor size may exceed the maximum penetration depth of light, making it less suitable for PDT. Therefore, the maximum propagation of the treatment light for PDT and the locations that can be reached with a light source determine which tumors can be treated with PDT. For this reason, tumor types commonly treated with PDT include superficial malignancies or disorders that are reachable using a flexible light source. These malignancies range from several cutaneous malignancies, including basal cell carcinoma and squamous cell carcinoma, head-and-neck as well as esophageal and lung cancers, to the bile duct, bladder, liver, colon, pancreatic, brain, ovarian, and prostate cancers [41,42].

### 3.5. Light Sources in PDT

Over the years, several different light sources have been used in PDT [43,44,45]. Preference for a certain light source depends on the location of the malignancy and the PS. Most frequently utilized sources constitute lasers and lamps, while recently, laser-emitting diodes (LEDs) are increasingly being employed [46,47]. Even the sun has been used as a light-source for PDT, in a variation that was termed daylight PDT [48,49]. Lamps generally contain a broad emission spectrum and allow illumination at a tunable wavelength in combination with optical filters. Lamps combined with the appropriate, preferably interchangeable, filters are therefore useable for multiple different PS that possess absorption maxima at corresponding wavelengths. They can produce light at sufficient intensity to induce photodynamic damage and are often manufactured in a portable size. However, lamps are usually not suitable for coupling to flexible fibers due to their poor beam quality, large beam size, and small power density. Therefore, they cannot be used to treat malignancies that require endoscopic light sources, e.g., fiber optics, restricting their use to certain superficial lesions. Conversely, lasers allow the coupling to flexible fibers and can therefore be used to treat more deeply seated tumors, such as malignancies of the colon. Due to their relatively high-power output, they produce sufficient light intensity at the tip of fibers required for photodynamic damage to tumor cells. Because lasers are mostly monochromatic, and many PS possess narrow absorption bands used for PDT, their emission wavelengths should be matched with the absorption maximum of the PS. For this reason, the use of a laser is restricted to a PS with absorption at or very close to the emission wavelength of the laser, which can be a disadvantage for clinics that employ the use of several different PS, requiring several expensive laser devices. Finally, LEDs are a relatively low-cost alternative to lasers that can be coupled to fiber optics [50]. They generally produce fluence rates of more than sufficient magnitude to allow for an efficient photodynamic effect.

### 3.6. The Importance of Fluence and Fluence Rate

The intensity of the light used in PDT is known as the fluence rate and is expressed as the output power over time per area or W/m^2^. Illumination at a certain fluence rate during a certain time gives rise to the total light dose known as fluence, expressed in J/m^2^. The impact of these factors was investigated by Henderson et al. in a study comparing several fluence rates for various fluences on mice bearing colon 26 tumors [51]. In general, low fluence rates were superior in their antitumor effect compared to high fluence rates both for antitumor efficacy as well as for inflammatory cytokine production in the tumor area. A fluence rate of 14 mW/cm^2^ was found to be optimal for tumor clearance in their setting, while animal survival dropped rapidly above 112 mW/cm^2^. In some animals, a lower fluence rate allowed a reduction in fluence for similar antitumor efficacy, demonstrating the importance of fluence rate. The enhanced antitumor effect of lower fluence rates was related to the availability of oxygen at the treatment site, which was rapidly depleted due to conversion into singlet oxygen at higher fluence rates of around 75 mW/cm^2^. At the lower fluence rates, oxygen was more available in the tumor area, possibly facilitating enhanced photodynamic damage. A more recent publication suggested that higher fluence rates may lead to increased EGFR activation, thereby contributing to a larger tumor burden compared to lower fluence rates [52], providing a possible partial explanation for the enhanced efficacy of lower fluence rates. Although this may be different in PDT settings using other sensitizers and therapeutic protocols, these data show that higher fluence rates are not necessarily more efficient in inducing photodynamic effect and cytotoxicity, which should be taken into consideration when designing PDT protocols.

### 3.7. Photodynamic Effect

Following the administration of a PS and irradiation with light, the energy of the photons can be absorbed by the PS, inducing an increased energetic state. This occurs only when the PS encounters light consisting of appropriate wavelengths, i.e., photons with an energy value that matches the available energetic states of the PS. Upon encountering such light, a PS may transition from its energetic ground state to the highly unstable singlet excited state (Figure 2). From there, it can emit the gained energy as heat which can also be used to induce cellular damage and/or fluorescence. In addition, the excited PS can move to a more stable triplet excited state through a process called intersystem crossing [53]. From there, the PS can transition back towards its stable ground state by phosphorescence, emission of heat, or by direct energy transfer to oxygen, yielding singlet oxygen (^1^O_2_), which is known as a Type II reaction [54,55,56,57]. Energy transfer may also occur to various types of biological molecules in a Type I reaction [54,55,56,57], forming radicals or radical ions. Of the possible outcomes, Type II is mechanistically simpler, and most PS are thought to engage in Type II rather than Type I reactions [54,55,56]. In any case, both reactions result in the production of ROS, which readily reacts with various (bio)molecules to irreversibly alter their structure, thereby inducing damage upon sufficient induction of ROS. The entire process of light absorption and PS-mediated transfer of the light’s energy to yield ROS is what constitutes the photodynamic effect in photodynamic therapy.

Jablonski diagram [58] illustrating type I and type II photoreactions of a PS after excitation with light. The energy of the light is absorbed, entering the PS to the singlet excited state (PS*). The singlet excited PS* can transition to its energetic ground state by releasing energy in the form of fluorescence or undergo intersystem crossing to enter the more stable excited triplet state. There, it can release its energy as a photon in the form of phosphorescence or by engaging in a type I reaction to yield ROS such as H_2_O_2_, OH^−^ and O_2_^−^, or in a type II reaction to yield singlet oxygen (^1^O_2_).

## 4. The Consequences of Photodynamic Therapy: Cell Death Pathways, DAMPs, ICD, Tissue Disruption, Vascular Disruption, and Immune Activation

The consequences of PDT are highly dependent on the location of the PS at the time of illumination. This is due to the short half-life of singlet oxygen molecules, decreasing the probability that it interacts with molecules further away from its site of origin [55,59,60,60,61]. For this reason, the location of a PS during illumination coincides with the type of cellular damage caused. Because PS displays preferences for different (sub)cellular locations [61], PDT can therefore facilitate the destruction of solid tumors in several ways. These can be further divided into direct consequences of photodynamic effect, including damage to tumor cells, to the tumor (micro)environment, and/or disruption of the tumor surrounding vasculature and indirect effects involving the induction of an innate (inflammatory) immune response (Figure 3) [41,62,63,64].

Photodynamic therapy occurs in the tumor after the administration of a photosensitizer and illumination with light. This results in direct damage to cells and structures in close proximity to the photosensitizer at the time of illumination, leading to disruption of the tumor vasculature followed by starvation of the tumor area and destruction of the tumor itself. Moreover, PDT can initiate immunogenic cell death, accompanied by the exposure and release of damage-associated molecular patterns (DAMPs). In this way, PDT can indirectly induce inflammation and infiltration of immune cells to the tumor and its vasculature.

### 4.1. Accidental Necrosis, Regulated Necrosis, Apoptosis, and Autophagy: Direct Damage to Tumor Cells

Direct antitumor effects of PDT-mediated damage include shrinking of the tumor mass through the initiation of various cell death pathways, including apoptosis, autophagy, accidental (non-regulated) necrosis, and several forms of regulated necrosis. Generally, it was noted that PS with preferential localization to mitochondria or membranes of organelles was prone to induce apoptosis, while PS that mainly localized to the plasma membrane was more prone to initiate necrosis [65,66,67,68,69]. It must be noted that the protocol used, i.e., the concentration, and the DLI, in addition to the fluence rate and fluence, can have a strong impact on the type of cell death initiated by PDT. Among the types of cell death initiated by PDT, necrosis is often induced by overwhelming photodynamic damage to the cell, leading to disruption of the structural integrity of the plasma membrane and a swift efflux of adenosine triphosphate (ATP) [70]. However, the exact sequential steps underlying PDT-induced necrosis are difficult to investigate and therefore remain elusive. However, the knowledge of the mechanisms of cell death, including regulated necrosis, is ever-expanding [71,72], and additional pathways, including necroptosis, are now also recognized to occur after PDT [73,74,75].

Apoptosis is a highly complex mode of cell death that is generally subdivided into extrinsic and intrinsic pathways. The intrinsic pathway may be triggered when the outer membrane of the mitochondrion is permeabilized, resulting in the activation of effector caspase 3 and 7. The extrinsic pathway is initiated through external factors that are detected at the plasma membrane, specifically involving caspase 8, eventually also resulting in the activation of effector caspases [76]. Apoptosis represents a more clearly defined mechanism of cell death in response to photodynamic damage compared to necrosis, especially the intrinsic pathway [65,67,68,77]. The Association of PS with mitochondria and subsequent PDT has been linked to the disruption of anti-apoptotic proteins, including members of the B-cell lymphoma 2 (BCL-2) family [19,78,79,80,81]. Conversely, upregulation of pro-apoptotic proteins, including BCL-2 associated X (BAX) at the mitochondrial membrane, with a corresponding release of mitochondrial proteins, including cytochrome c, was shown to occur after PDT [78,82,83,84,85,86,87,88]. Mitochondrial membrane permeabilization in response to PDT is most likely facilitated through the opening of the mitochondrial inner membrane pore. This is evidenced by the observation that the loss of membrane potential, resulting in apoptosis after photodynamic damage, is abolished following transfection with anti-apoptotic Bcl-2 [89]. Moreover, this is also evidenced by the affinity of PS for mitochondrial membranes associated with the regulation of such pores [89,90,91], by the inhibition of photocytotoxicity and cytochrome c release when introducing inhibitors of inner membrane pore permeabilization [92] and by additional studies reporting permeabilization after photodynamic damage [89,92,93,94,95]. Furthermore, PDT has been shown to induce increased expression of Fas and FasL in vitro and in vivo [96,97], indicating the involvement of the extrinsic pathway of apoptosis. Finally, PDT-induced release of mitochondrial apoptosis inducible factor (AIF) has been shown [91,98,99], suggesting a role for caspase-independent apoptosis induction after PDT.

Autophagy is a catabolic process that can serve as either a pro-survival or death mechanism, depending on the location and amount of damage to the cell [100]. The process involves the formation of a lipid bilayer membrane vesicle that envelops damaged cellular components and the subsequent integration of this vesicle into the lysosomal system, thereby forming the autolysosome, followed by degradation of the engulfed content. The detection of autolysosomes in dying cells is not an indication that autophagy was the cause of death but could rather indicate attempts for cellular rescue [101,102]. Evidence indicates that PDT-induced autophagy can function as a death mechanism, especially above a certain threshold of cellular damage [85,103,104,105]. Photodynamic damage to organelles or cytosolic proteins induces damage that, upon irreversible oxidation, can also potentially initiate autophagy [100]. Moreover, markers of autophagy were elevated after PDT, including conversion of the LC3-I protein to the active form LC3-II and clustering of labeled LC3-II [106]. Silencing of the key autophagy protein Atg7 was shown to increase apoptosis after low-dose PDT versus control cells [107,108], indicating a cytoprotective role for autophagy in this context. Conversely, the knockdown of Atg7 increased resistance to PDT-induced cell death in MCF-7 cells [109]. These results suggest a differential role for Atg7 in susceptibility to PDT for separate cell lines. In apoptosis-hampered *Bax^−/−^* cells [105,110] or *Bax^−/−^*/*Bak^−/−^* cells [85], autophagy was shown to play a key role in PDT-induced cell death, as *Bax^−/−^* cells did not display hampered cell death and showed increased vacuolar morphology compared to *Bax^+/+^* cells following PDT. These results indicate the importance of autophagy in such cells. For all PS reported to be involved in autophagy, the settings, as well as the tumor model investigated, appear to influence the outcome of autophagy in which more photodynamic damage appears to tilt the balance toward cytotoxicity and less PDT-induced damage toward cytoprotection.

### 4.2. Damaging the Tumor Vasculature

Another type of antitumor mechanism initiated by PDT involves damage to the vasculature present in and around the tumor [41,111,112,113,114]. Several PS, including Verteporfin, TOOKAD, and NPe6, are known to preferentially remain in the vasculature after intravenous administration and can therefore inflict severe damage to the tumor vasculature after illumination of the tumor area [115]. In vitro, reports have shown that endothelial cells are more sensitive to photodynamic damage than fibroblasts and smooth muscle cells, in spite of the similar accumulation of PS across these cell types [116]. Furthermore, it was found that endothelial cells displayed vastly increased PS accumulation and enhanced sensitivity to PDT compared to human colon adenocarcinoma cells with a similar proliferation rate, providing a rationale for targeting endothelial cells in vitro [117]. In the preclinical stage, strategies that focus on disrupting the tumor vasculature have been investigated extensively [118,119]. This type of treatment was termed vascular-PDT and has shown efficacy and safety in trials for prostate cancer [120,121].

In recent years, various molecules in the tumor vasculature have been targeted for vascular-PDT to enhance specificity, often by linking PS to carriers, including vessel-targeted antibodies or nanoparticles. Targets include CD13, CD276, extra domains of fibronectin, integrin αvβ3, neurophilin-1, nucleolin, platelet-derived growth factor receptor β, tissue factor, and vascular endothelial growth factor receptor 2 [115]. Although some have been studied more intensively than others, it is difficult to predict the optimal target for vascular PDT. Many PS used for vascular PDT induce strong vascular disruption and shutdown without targeting agents, bypassing the need for such targeting moieties. Of note, there are some indications of PDT-induced changes in vascular permeability. In line with this, several studies using different PS and tumor models show increased permeability after PDT treatment [122,123,124,125]. The implications of this observation for drug delivery have been evaluated by investigating the potential for PDT to enhance the accumulation of, e.g., antineoplastic agents in the tumor. In line with this, PDT was found to enhance the accumulation and antitumor efficacy of Doxil (liposome-encapsulated doxorubicin) while reducing doxorubicin-related side effects [123,126,127]. Moreover, our group has recently shown by intravital microscopy that Radachlorin PDT induces complete vascular disruption in the tumor area while vessels surrounding the PDT area remain intact (Section 3) [114]. In the same setting, nanoparticles displayed enhanced accumulation after PDT and showed preferential uptake in immune cells of myeloid origin. Together, these studies show the safety and efficacy of vascular-PDT and indicate the potential of strategies that combine PDT with carrier-based modalities to enhance the accumulation of their encapsulated agents at the tumor site, thereby potentially enhancing treatment outcomes while reducing side effects.

### 4.3. Damage-Associated Molecular Patterns and Immunogenic Cell Death

Classically, apoptosis has been considered to be immunologically silent and possibly tolerogenic, while necrosis was associated with inflammation. This was based on the notion that apoptotic cells or bodies are phagocytosed, preventing the release of intracellular content with immunostimulatory activity, whereas necrosis results in the release of such intracellular molecules [20]. These immunostimulatory molecules derived from dying cells are known as damage-associated molecular patterns (DAMPs), which are normally shielded from immune cells in intact, ‘healthy’ cells. Upon cell death, however, they may be exposed or released, allowing binding to immune cells through pattern-recognition receptors (PRRs) [128], including Toll-like receptors (TLRs), RIG-like receptors, and NOD-like receptors [21,29]. Chronic exposure of DAMPs to these PRRs can trigger inflammation, and any type of cell death that can elicit an immune response is called immunogenic cell death (ICD) [129]. With the discovery of various modes of cell death, it has been noticed that apoptosis, accidental necrosis, regulated necrosis, and occasionally autophagy can all trigger the exposure and/or release of DAMPs and thus induce ICD [24,32,33,130]. Several types of DAMPs that are important for ICD currently include high-mobility group box 1 (HMGB1), calreticulin (CRT), adenosine triphosphate (ATP), and heat-shock proteins (HSPs). PDT with numerous types of PS has been shown to lead to exposure or release of DAMPs [22,27,28,29,30,31,64,131,132,133,134,135,136,137,138,139,140]. The degree and the type of DAMP exposure are shown to be dependent on the PS as well as the tumor model used and can therefore vary between studies [141]. Of the DAMPs induced after PDT, HMGB1 is present in the nucleus under normal conditions but is released into the extracellular matrix after induction of ICD, where it functions as an inflammatory cytokine that mainly binds to TLR4 and receptor for advanced glycation end products (RAGE) [142]. HSPs are chaperones important for protein folding and sensing as well as correctly re-folding misfolded proteins, thereby playing a pro-survival role. However, during ICD, certain HSPs are translocated to the cell surface and induce phagocytosis of the cell by antigen-presenting cells (APCs) or trigger recognition by natural killer (NK) cells through CD94 [130]. The induction of ICD can also result in the extracellular release of ATP, where it functions as an APC-attracting and inflammasome-activating agent by interacting with purinergic P2Y2 or P2X7 receptors (P2RY2 or P2RX7), triggering the release of the proinflammatory cytokine IL-1β [143,144]. The release of ATP can occur after the loss of structural integrity of the plasma membrane but also independently of plasma membrane permeabilization through the classical, protein kinase R (PKR)-like endoplasmic reticulum kinase (PERK)-regulated, proximal secretory pathway and PI3K-dependent exocytosis [31]. The mechanism behind ATP release during ICD is complex and depends on the type of ICD inducer and apoptotic stage of the cell [29]. Another DAMP, CRT, is translocated from its most prominent location, the lumen of the endoplasmatic reticulum (ER), to the cell surface, where it can be recognized by CD91 on professional APCs. This may trigger phagocytosis and subsequent processing of potential (neo)antigens for cross-presentation to T cells, thereby possibly initiating adaptive immune responses [22,145].

### 4.4. Antitumor Immune Responses

The efficiency with which PDT induces ICD with DAMP exposure and release may lead to acute inflammation in the tumor microenvironment that can initiate an antitumor immune response. The early phase of this immune response consists of tumor-infiltrating lymphocytes (TILs), primarily of myeloid origin, that attempt to neutralize the PDT-induced damaged cells, cellular debris, and released content. Subsets involved in this early response were shown to include neutrophils [146,147,148], accompanied by a systemic increase in IL-6 [146,149]. These neutrophils were also temporarily increased in the tumor-draining lymph nodes (dLN) in addition to the tumor [150]. It was shown that photodamage to the tumor vasculature induces contraction of endothelial cells, thereby allowing the adhesion of neutrophils through β2 integrin receptors [148,151]. Blocking of neutrophil entry into the dLN was shown to reduce the number of activated CD8^+^ T cells and the efficacy of PDT, indicating the importance of this process [144,150,152]. PDT-induced damage was shown to result in macrophage activation and TNFα production through stimulation of TLR2/4 [28]. Moreover, macrophage-related clearance of the newly formed damage was shown to occur after PDT [153]. Natural killer (NK) cells were also shown to be involved in the immune response after PDT, as control of distant disease by CD8+ T cells was reported to be enhanced by NK cells [154]. However, the exact role of NK cells in this process remains to be elucidated. Although dendritic cells (DCs) are often cited as crucial for the PDT-induced antitumor immune response through phagocytosis and cross-presentation of tumor (neo)antigens, their exact role in this process remains elusive. However, PDT-induced dying cancer cells have been shown to induce the maturation of DCs [131,155,156], a process that was hampered by the neutralization of DAMPs [157].

Theoretically, PDT-induced cell death can facilitate the release of previously inaccessible cancer (neo)epitopes that can subsequently be phagocytosed, processed, and cross-presented by DCs to mount a tumor-specific T cell response when in the presence of danger signals such as PRR-ligands including DAMPs (Figure 4). In this way, PDT functions as an in-situ vaccination strategy to induce a tumor-specific adaptive immune response while debulking the tumor and/or disrupting the tumor vasculature. To show the feasibility of this strategy, vaccination studies that injected PDT-treated dying tumor cells into animals to monitor the immune response and tumor size after treatment have been performed and often showed increased numbers of CD8^+^ T cells in the tumor and a treatment-induced tumor growth inhibition [158,159,160,161]. Moreover, naïve DCs injected intratumorally after PDT were shown to capture tumor antigens, mature, migrate to draining lymph nodes and potentiate tumor-specific T cells [162]. Some studies that employ PDT as an in-situ vaccination strategy show, in particular for vascular-targeted PDT, enhanced activation of tumor-specific antitumor CD8^+^ T cells against cancer cells expressing exogenously introduced antigens [152,163,164], HPV E6/7 oncoproteins [165], cancer/testis antigen P1A [166], and murine colon 38 (MC38) neoepitopes [131]. The induction of such adaptive immune responses, in turn, was found to be essential for the antitumor efficacy observed after PDT [167]. In line with this, depletion of CD8^+^ T cells was shown to drastically reduce tumor growth inhibition and to enhance progression-free survival induced by PDT [131,154,156,165,168], while CD4+ T cells supported PDT efficacy [169,170]. Furthermore, in immunodeficient mice, in which a strong long-term antitumor response is seriously diminished or even absent [171,172], the subsequent transplantation of naïve T cells before PDT, but not after, from immunocompetent mice induced tumor growth delay [172]. Adoptive transfer of bone marrow cells from immunocompetent mice induced an even stronger antitumor effect in this setting. Interestingly, in a PDT-treated patient with tumors on the upper right limb, additional untreated tumors on the same limb went into remission and remained so over 15 months, indicating a potential distant (abscopal) effect [173]. Biopsies of the illuminated tumors taken a month post-treatment revealed strong infiltration of CD8+ T cells. In addition, it was found that patients with vulvar intraepithelial neoplasia unresponsive to aminolevulinic acid (ALA)-PDT were more likely to downregulate major histocompatibility complex (MHC)-I expression [174]. As CD8^+^ mediated tumor cell lysis is dependent on a proper cell surface expression of MHC-I molecules [175,176,177], this presumably contributes to an impaired immune-mediated tumor clearance after PDT. The potential of PDT to induce an abscopal effect on distant tumors not treated by PDT through the induction of antitumor immune responses has been shown in several reports [131,168,178].

Schematic representation of PDT-induced tumor-specific immune responses. (1) PDT induces cell death in the tumor, releasing DAMPs and previously inaccessible (neo)epitopes. (2) Immature DCs phagocytose the DAMPs, (neo)epitopes, and tumor debris, inducing (3) maturation and (4) migration to secondary lymphoid organs. (5) In those secondary lymphoid organs, such as lymph nodes, the (neo)epitopes are cross-presented to T cells, initiating tumor-specific T cells. (6) The T cells expand and migrate from the lymph node (7) to the PDT-treated tumor and metastases to induce antitumor immunity.

Together, these observations suggest that PDT could be complemented with immunotherapy against cancer, including immune checkpoint inhibition (ICI), to improve the therapeutic outcome on both primary (PDT-treated) and distant (not treated with PDT) tumors. In line with this, several immunotherapies, including checkpoint blockade antibody treatment [179,180,181], nanoparticle-encapsulated TLR-ligands polyinosinic: polycytidylic acid (poly I:C) with R848 in addition to lymphocyte-attracting agent CCL20 [131], CpG [182,183], and an indoleamine 2,3-dioxygenase inhibitor [184] and cancer-specific therapeutic vaccination protocols [168] have been shown to enhance PDT against primary and distant tumors. These findings strongly indicate the potential for PDT to induce adaptive antitumor immune responses, which are required for direct PDT-mediated antitumor effects as well as for long-lasting protective antitumor immunity.

In summary, the antitumor mechanisms of PDT consist of (I) direct damage to cells and structures in the tumor area, (II) damage to the tumor vasculature leading to vessel disruption or collapse, which leads to tumor destruction, and (III) the induction of cellular antitumor immune responses through PDT-induced immunogenic cell death. These mechanisms can occur simultaneously and explain the effective and systemic antitumor efficacy often observed after PDT.

## 5. Current Challenges and Opportunities of Photodynamic Therapy against Cancer

The most notable challenges for PDT (Figure 5) include: (I) undesirable distribution of PS after intravenous administration, (II) attenuation of light through tissues that can result in incomplete light delivery to the tumor area, (III) hypoxia in the tumor environment that depletes the oxygen available for PDT, (IV) incomplete or transient tumor vasculature disruption after PDT and subsequent angiogenesis or vessel repair, (V) partial tumor destruction after PDT followed by tumor relapse, or (VI) insufficient PDT-mediated induction of antitumor immune responses. These efficacy-reducing factors and recent efforts for improvement, either by enhancing PDT or by combination with other modalities, will be addressed.

The current challenges of PDT: (I) unfavorable biodistribution of PS, (II) limited light propagation through tissues, (III) reduced number of ROS through hypoxia in the tumor, (IV) temporary vascular disruption followed by repair, (V) partial tumor destruction followed by tumor relapse, (VI) insufficient induction of antitumor immune responses.

## 6. Biodistribution of Photosensitizers

A longstanding challenge of PDT is related to the poor tumor distribution that many PS tend to present after administration, reducing the accumulation of PS in the tumor and potentially hampering the complete diffusion of PS throughout the tumor. This may result in a reduced number of active molecules in the tumor area as well as throughout the various tumor layers and induce incomplete destruction of the tumor area. Furthermore, unfavorable biodistribution may facilitate prolonged periods of unwanted retention in non-target tissues. This is known as photosensitivity and often affects patients when substantial amounts of PS remain in the skin. For these reasons, research has focused on improving the distribution of PS to enhance antitumor efficacy and reduce off-target accumulation. This is often achieved by employing carrier systems, including antibodies, liposomes, and nanoparticles (NPs), that may facilitate a more favorable biodistribution. Encapsulation in or association with these carrier systems introduces a degree of control in the distribution of the PS-carrier complex. Several carriers that have been employed to improve the biodistribution of PS will be discussed.

### 6.1. Nanoparticles

Many nanoparticles (NP)-based carriers have been created and characterized in vitro [185,186,187,188,189,190,191,192]. Two studies reported the use of 9-fluorenylmethoxycarbonyl-L-lysine that encapsulated chlorin e6 (Ce6) [193] and gold NPs carrying a derivative of the poorly soluble Zn(ii)-phthalocyanine and polyethylene glycol (PEG) [194], respectively. Both particles accumulated in tumors over time and induced tumor growth inhibition after PDT in murine models. In another paper, a carrier consisting of polyhedral oligomeric silsesquioxane was PEGylated and crosslinked with Ce6 [195]. Distribution to xenografted U14 murine cervical tumors was higher compared to free Ce6. Furthermore, the tumor accumulation of the particles was higher in the tumors compared to the heart, liver, lung, spleen, and kidneys. PDT with the particles induced a tumor growth inhibition that was significantly better than PDT with free Ce6, showing the potential of carriers for PS. The PS PpIX was conjugated to PEGylated glycol chitosan to form NPs that display increased tumor accumulation after i.v. administration compared to its non-PEGylated equivalent and free PpIX, with fluorescence intensity in the tumor being increased compared to the heart, liver, spleen, lung, and kidneys [196]. In addition, PDT with the NPs induced significantly enhanced tumor growth inhibition versus its non-PEGylated version as well as PpIX alone.

Several studies investigated the use of NPs that require activation before treatment, functioning as pre-drug carriers for PS that introduce a measure of controlled release. For example, Ce6 was incorporated into PEGylated NPs, either alone [197] or with chemotherapy [198], that disintegrate at acidic environments encountered inside cells and in which Ce6 was quenched before disintegration. These particles displayed enhanced accumulation in tumors compared to free Ce6, and PDT with the particles showed an increased tumor growth inhibition compared to free Ce6. In another study, Ce6 was encapsulated in NPs that consisted of imidazole that disintegrated in response to ^1^O_2_ to allow control of Ce6 release [199]. These NPs accumulated in tumors after i.v. administration at increased amounts compared to free Ce6 significantly enhanced tumor growth inhibition for the ^1^O_2_ responsive micelles compared to ^1^O_2_ unresponsive micelles, to free Ce6 and untreated. In another approach, Ce6 was incorporated into NPs together with a near-infrared (NIR) dye in which Ce6 was quenched until the NPs were activated by photobleaching the NIR dye, allowing photo-activation of the NPs [200]. In mice, the NPs distributed to tumors over time and treatment with the NPs led to significantly enhanced tumor growth inhibition compared to the particle without photobleaching and compared to NPs without the NIR dye in which Ce6 was not quenched. The authors also report that edema was reduced at locations where no photobleaching was performed compared to a non-quenched version of the NP, indicating the ability of the activatable NPs to reduce off-target effects. When taken together, these results show the potential of (activatable) NPs for enhancing the tumor distribution of PS, both for increased therapeutic effect and reduced retention in off-target organs.

### 6.2. Antibodies for Tumor Targeting

Antibodies are a frequently studied carrier for PS with a high degree of selectivity as a result of antibody-determined specificity. Several antibody-PS conjugates were created and evaluated in vitro [201,202,203]. In the preclinical setting, Benzoporphyrin Derivative (BPD) was conjugated to the EGFR-blocking antibody Cetuximab (Cet-BPD) [204,205,206]. The BPD was linked in a ratio that allows the quenching of fluorescence and partial deactivation of phototoxicity. The results indicate that Cet-BPD is internalized, such as Cetuximab, into the lysosome as part of the EGFR trafficking pathway, where it dissociates, thereby restoring BPD’s fluorescence and phototoxicity. In a mouse model, BPD-Cet displayed increased tumor selectivity compared to free BPD (Verteporfin) [207] and was found to increase the maximum tolerated photodynamic dose up to ~17 times compared to free BPD. Administration of Cet-BPD without illumination decreased metastatic burden compared to control in nude mice with orthotopic EOC tumors, characterized by disseminated metastases, and PDT treatment further reduced metastatic burden, which was significantly enhanced in conjunction with paclitaxel. Together, these data show that Cet-BPD can be used as an efficient antitumor modality. The PS IRDye700DX was conjugated to human anti-human C2-45 antibody that is specific for carcinoembryonic antigen (CEA) [208]. The conjugate, termed 45IR, displayed CEA-specific binding to cells and induced strong CEA-specific cell death after illumination with NIR light in various CEA-positive cells. PDT with 45IR was found to effectively inhibit tumor growth in murine models, showing its potential in a preclinical setting. The PS IRDye700DX dye was also conjugated to basiliximab and panitumumab, which was administered as a cocktail for PDT, inducing a growth inhibition in tumor-bearing mice [209]. In another study, Ce6 was conjugated to trastuzumab, an antibody specific for the HER2 antigen [210]. This conjugate displayed a 6-fold increase in the accumulation of tumors when compared to free Ce6, showing the enhanced distribution. Finally, the anti-EGFR antibody panitumumab was coupled to IRDye700DX and investigated in a postsurgical setting to remove microscopic tumor remnants [211]. PDT with this conjugate showed significantly enhanced tumor control after the removal of 50% and 90% of the tumors compared to resection only and administration of the conjugate without illumination. Together, these results show the benefit of antibodies as carriers for PS to enhance tumor accumulation. Moreover, it shows that antibody-PS conjugates can display a dual mode of action, whereby the therapeutic potential of the antibodies is combined with PDT in a single agent.

### 6.3. Peptides

Peptides have also been used as carriers for PS, including Tat (a cell-penetrating peptide) [212], albumin [213], and biotin [214], that have been evaluated in vitro. For in vivo studies, folic acid was conjugated to the PS pyropheophorbide A, displaying enhanced tumor accumulation tumors that express folate receptor versus free pyropheophorbide a, but much less in tumors that do not express the folate receptor [215]. In mice, PDT with the conjugate induced complete regressions of folate receptor-expressing tumors, providing a proof-of-concept for the approach. A different study employed the αvβ6 integrin-targeting peptide, designated as HK, that was functionalized with graphene coated with the PS HPPH [216]. The accumulation of the construct in tumors was shown to be increased compared to free HPPH or PEGylated constructs without the HK peptide. PDT with the construct resulted in significant tumor growth inhibition as well as increased expression of CD70 and CD40 in CD11c^+^ DCs in tumor-draining lymph nodes and enhanced IFN-y levels in serum, suggesting that the treatment induces an antitumor immune response. Recently, virus-like particles (VLPs) derived from human papillomavirus (HPV) were shown to preferentially bind to several cancer over non-cancer cell lines [217] by binding to Heparan sulfate proteoglycans [218,219,220]. Based on this, these HPV-based VLPs were conjugated to the PS IRDye700DX, which was intended for the treatment of primary uveal melanoma. The resulting PS, AU-011, was shown to induce necrosis in uveal melanoma xenografts in rabbits [221]. Expanding on this, AU-011 was combined with checkpoint blockade antibodies anti-CTLA-4 and anti-PL-L1, inducing complete or near-complete responses, respectively, in TC-1 tumor-bearing mice [222]. These results indicate that peptides hold great potential as carriers for PS by enhancing their tumor accumulation and improving antitumor efficacy after treatment.

### 6.4. Extracellular Vesicles

Recently, extracellular vesicles (EVs) were investigated as drug carriers for PS. EVs are nanosized cell-derived vesicles and are involved in communication between cells, as well as in immunological processes such as antigen presentation [223]. They consist of a lipid-bilayer with several membrane proteins and can encapsulate various types of lipids, proteins, and genetic information, including DNA, mRNA, lncRNA, and microRNA [224,225,226,227,228]. In cancer treatment, they were reported to be promising carriers for microRNAs [229,230], doxorubicin [231], and paclitaxel [232]. Moreover, several studies investigated EVs as carriers for the PS meta-tetra(hydroxyphenyl)chlorine (mTHPC) [233,234]. They show enhanced tumor accumulation and improved tumor growth inhibition of the mTHPC-EVs compared to liposomal formulations of mTHPC (Foslip) or free mTHPC. In a follow-up study, mTHPCs-EVs were shown to induce infiltration of immune cells into murine tumors, but this did not result in complete rejection of the infiltrated tumors [235]. In a different study, M1-like macrophage-derived EVs were used as carriers for Ce6 and a pH-responsive prodrug version of doxorubicin [236]. These EVs were able to polarize M2-like macrophages to an M1-like phenotype and induce strong antitumor effects in tumor-bearing mice, significantly enhancing survival compared to all relevant controls. Another approach used EVs as a carrier for Zinc Phthalocyanine (ZnPc), a potent but poorly soluble PS [237]. The ZnPc-EVs displayed a preferential uptake in cancer cells over immune cells In vitro and were shown to accumulate in tumors after administration in mice. PDT with the ZnPc-EVs induced a strong tumor growth inhibition without inducing notable off-target toxicities. Together, these data show the potential of EVs as carriers for PS in PDT treatment.

### 6.5. Sensitizer-Loaded Immune Cells

In a more experimental study, gold nanoparticles were crosslinked to Ce6 molecules, conjugated to NHS-PEG-aCD3 antibodies, and loaded into cytokine-induced killer (CIK) cells as a delivery system [238]. CIK cells are purified human PBMCs that are cultured in the presence of IFN-y, anti-CD3, and IL-2 before further use. In mice, the Ce6-NHS-PEG-aCD3-CIK cells showed enhanced accumulation in tumors compared to Ce6-NHS-PEG-aCD3 not loaded in CIKs. Unloaded CIK alone induced a tumor growth inhibition. However, this was significantly enhanced after PDT with Ce6-NHS-PEG-aCD3-CIK. This study shows that immune cells can also be used as carriers for PDT.

## 7. Light Propagation through Tissues

The attenuation of light in tissues limits the maximum tissue penetration of light. This may result in incomplete illumination of the tumor area if the depth of the tumor exceeds the maximum light penetration used for PDT (reviewed in [41,239]). This depth is usually several millimeters, up to approximately a centimeter into the tissue measuring from the surface of illumination, but is highly dependent on the wavelength used and the properties of the tissue. For PDT in patients, this can result in the partial destruction of the tumor and its surrounding vasculature, leading to tumor regrowth. Reducing the attenuation of light through tissues to optimize PDT involves a tradeoff, whereby wavelengths in the deep red or NIR spectrum generally penetrate further into tissues but contain reduced energy compared to smaller wavelengths for the generation of ROS. This can, fortunately, be compensated by increasing the fluence but must be performed in a manner that prevents tissue heating as a consequence of the high energy of the light.

### 7.1. NIR-Absorbing Sensitizers

Several PS that allow absorption in the NIR region have been under recent investigation. In line with this, novel PS consisting of black titania [240] or conjugated polymer nanoparticles (CP-NPs) [241] were shown to produce heat for photothermal therapy (PTT) and ROS for PDT after illumination with NIR light. Both PS were distributed to the tumor area and induced a tumor growth inhibition in tumor-bearing mice. A different NIR-light absorbing molecule consists of NaYbF_4_ NPs [242], which were shown to significantly inhibit tumor growth after intratumoral administration followed by PDT. Luciferase-expressing cells were engrafted inside the tibias of nude mice as a model for bone metastasis and treated with PDT after injection of the NPs into the bone matrix. Tumor growth inhibition was also observed in this setting compared to the control, indicating that more deep-seated tissues can be treated with PDT using these NPs. A new kind of green titania with absorption in the NIR region was conjugated to triphenylphosphonium (TPP) to target mitochondria [243]. PDT treatment with NIR light on a single tumor in mice bearing two tumors on opposite flanks significantly inhibited tumor growth in the treated tumors compared to the internal control tumor or versus animals treated with NIR light alone, the PS alone, or untreated animals. Of note, the clinically approved indocyanine green (ICG) is a fluorescent dye that was investigated extensively in the preclinical setting due to its diagnostic and therapeutic properties [244,245,246,247,248,249], but its instability in the body has hampered its use in the clinic. To this end, an encapsulated form of ICG was investigated in patients, showing an improved clinical response against basal cell carcinoma [250]. Another sensitizer, a bacteriochlorin named redaporfin [251], was recently characterized and investigated for its potential in vascular-targeted PDT, showing strong antineoplastic activity in murine models through deprivation of the blood supply [252]. Redaporfin-PDT was shown to induce immunogenic cell death by selective destruction of the endoplasmic reticulum and the Golgi apparatus [253]. Moreover, vascular-PDT with redaporfin was shown to induce infiltration of neutrophils, a systemic increase in IL-6, increased levels of IFN-ƴ-producing or CD69^+^ CD4^+^ and CD8^+^ T cells, and an increased CD4^+^/CD8^+^ T cell ratio [146]. The same study reports that the therapeutic effect of redaporfin-PDT was dependent on neutrophils and CD8^+^ T cells but not CD4^+^ T cells. In the clinic, redaporfin has shown high efficacy in combination with checkpoint blockade antibody treatment in a case report against head and neck cancer [254] and has been in phase I/II trial for the same condition (NCT02070432). Together, these approaches attempting to enable the use of higher wavelengths for optimal tissue penetration are promising.

### 7.2. Upconversion Nanoparticles

Another approach involves the utilization of upconversion nanoparticles (UCNPs) that absorb multiple photons at a certain wavelength and converts these into a photon through an anti-Stokes shift with shorter wavelengths that therefore contains more energy and can be used to excite a PS for PDT [255,256]. As a proof of concept, many UCNPs were created and characterized [257,258,259,260,261,262,263,264]. For in vivo application, Gao et al. designed UCNPs that were loaded with ZnPc as a PS and conjugated to c(RGDyK) to target the vasculature surrounding the tumor [127]. The UCNPs displayed enhanced accumulation after administration in the tumor area compared to UCNPs lacking c(RGDyK). PDT with this UCNP induced enhanced vascular permeability in the tumor area, which subsequently increased the tumor accumulation of injected particles. In order to test the efficacy of the UCNPs, 1 cm pork tissue was placed between the tumor and the light source, after which PDT was combined with the chemotherapeutic agent Doxil. This treatment regimen resulted in a significantly enhanced tumor inhibition rate of the modified UCNP versus Doxil alone, showing that the UCNPs enabled PDT in deep-seated tissues. In other studies, Ce6-loaded [265,266] UCNPs and AgBiS_2_-loaded [267] UCNPs were created, inducing significant tumor growth inhibition after PDT at high wavelengths for upconversion. Similarly, UCNPs were created that also contained doxorubicin for chemotherapy [268]. PDT with the particles administered intratumorally induced tumor growth inhibition but did not provide a clear benefit over UCNPs that do not include doxorubicin. When taken together, these data indicate the potential of approaches that attempt to overcome the limitations of light propagation through tissues.

## 8. Hypoxia in the Tumor Area

The hypoxic environment that is commonly observed in solid tumors can be a challenging factor for the efficacy of PDT. Hypoxia is defined by limited availability of O_2_, the main acceptor for the energy transfer of light, resulting in a small available amount at the site of action. This can reduce photodynamic damage through singlet oxygen generation and, therefore, diminish the therapeutic effect. Recent efforts to address hypoxia in the context of PDT involve the use of PS that does not require large amounts of oxygen for its function. For example, NIR-absorbing gold nanorods that induced plasmonic heating upon illumination were functionalized with endoperoxides that can induce the release of singlet oxygen as a result of plasmonic heating. In this way, the PS functions as its own source of oxygen, thereby circumventing hypoxia [269]. Other sensitizers were developed to remain cytotoxic under hypoxic conditions In vitro and await further testing in vivo [270,271].

### 8.1. Sensitizers That Function As Their Own Source of Oxygen

One study used nanocomposites that facilitate the light-dependent splitting of water molecules to generate oxygen and contain the PS PpIX to consume the oxygen molecules for PDT [272]. This nanocomposite was shown to increase singlet oxygen generation In vitro under hypoxic conditions compared to free PpIX, as well as induce significant 4T1 cell viability reduction in both hypoxic and normoxic conditions. In contrast, PpIX-PDT with the same light regimen only resulted in pronounced cell death in normoxic, but not hypoxic, conditions. Moreover, the construct was shown to distribute to murine tumors, and PDT in tumor-bearing mice induced an enhanced growth inhibition compared to PpIX alone, showing the potential of the approach. In another study, Ce6-loaded manganese ferrite nanoparticles were created that can catalyze the H_2_O_2_ present in the tumors to generate O_2_ for PDT [273]. These particles also strongly reduced cancer cell viability under normoxic as well as hypoxic conditions. In tumor-bearing animals, the particle reduced the level of hypoxia in the tumor and induced an improved tumor growth inhibition after PDT compared to all relevant controls. One study loaded nanoparticles that contain perfluorocarbon (PFC) as an O_2_ carrier and ICG as a PS into red blood cell membranes [274]. The resulting particles enhanced accumulation in tumors compared to the NPs not loaded into red blood cell membranes. Tumor-bearing mice treated with PDT showed an improved tumor growth inhibition for the particle compared to all relevant controls without inducing cures. Similarly, the PS IR780 was loaded into nanodroplets containing PFC with an O_2_-loading capacity to alleviate tumor hypoxia [275]. The construct induced cancer cell death under hypoxic conditions and PDT in tumor-bearing mice induced significant tumor growth inhibition versus all relevant controls. These data show the feasibility of sensitizers that are their own source of oxygen. Another study created metal-organic frameworks (MOFs) containing molecules that convert H_2_O_2_ to O_2_ and a sensitizer for PDT [276]. The MOFs strongly reduced cancer cell viability In vitro, although the dark toxicity of the particles was not assessed. Treatment with the MOFs in U14 tumor-bearing mice induced a tumor growth inhibition and increased survival rates.

### 8.2. Hypoxia-Responsive Prodrugs

Another approach incorporated ICG into iRGD-modified nanoparticles together with the hypoxia-activated prodrug tirapazamine (TPZ), using tumor hypoxia to convert TPZ to its active form [277]. The particle-induced cancer cell death under both normoxic and hypoxic conditions, whereby TPZ specifically improved the cell death of the NPs under hypoxic conditions. In mice, the tumor distribution of the particle, through the inclusion of iRGD, was increased compared to free ICG. PDT with the NPs significantly improved tumor growth inhibition compared to all relevant controls. Another study employed the use of hypoxia-responsive prodrug TPZ, which was co-encapsulated with ICG in PEGylated PLGA-based NPs and conjugated with iRGD [277]. These particles induced a strong reduction in cancer cell viability under normoxic and hypoxic conditions In vitro. The particles accumulated in murine tumors to a much larger degree than the particles lacking iRGD or free ICG, as well as compared to other organs. In tumor-bearing mice, PDT with the particles induced a statistically significant tumor growth inhibition, and reduction in lung metastasis after treatment was compared to all relevant controls. A different approach involved the use of UCNPs functionalized with the PS Rose Bengal (RB) for RB-PDT and a hypoxia probe installed on the surface of red blood cells with folic acid inserted into the membrane for a measure of tumor selectivity [278]. In hypoxic environments, the hypoxia probe can release O_2_ release from oxygenated hemoglobin upon illumination, strongly reducing cancer cell viability In vitro. The folic acid addition enhanced the tumor accumulation after administration, and PDT with the UCNPs enhanced tumor growth inhibition compared to the relevant controls. These studies show the potential of using hypoxia-responsive drugs for PDT.

### 8.3. Diffusion of Oxygen in the Tumor

In another study, hyaluronidase was used to disrupt the extracellular matrix in the tumor to enhance the diffusion of oxygen and, potentially, the efficacy of nanoparticle-mediated PDT [279]. Biodistribution data showed a 2-fold increase in tumor accumulation when treated with intratumoral hyaluronidase before administration of the NPs. Treatment with hyaluronidase also increased the oxygen content throughout the tumor and enhanced the tumor growth inhibition induced by PDT versus the same treatment without hyaluronidase, indicating the potential of extracellular matrix modeling prior to PDT.

Together, these data show the potential of PDT-enhancing modalities that alleviate or circumvent the hypoxic environment often observed in tumors.

## 9. Vascular Disruption

Photodynamic therapy can be performed with the intention of disrupting the vasculature by damaging the endothelial layer of the tumor vasculature, often by employing types of PS (e.g., verteporfin, padoporfin, and padeliporfin) that mostly retain in the vasculature after administration. This type of PDT is called vascular-targeted PDT (VTP) and has been shown in a preclinical setting to be an effective antitumor approach with excellent safety profiles [115,146]. However, incomplete or temporary vessel shutdown, in addition to angiogenesis that restores the tumor vasculature, are the main factors that result in relapse after treatment [280,281,282]. In order to address this, several strategies have been under recent investigation.

### 9.1. Tumor Vasculature Disrupting Agents

To improve treatment outcomes, VTP can be combined with compounds that further disrupt or ‘normalize’ tumor vessels to effectively treat the tumor. One example of this approach utilized inhibitors of the phosphatidylinositol 3-kinase (PI3K) pathway in combination with verteporfin-PDT [283], as the PI3K pathway was shown to promote endothelial cell survival and proliferation after PDT treatment. The study identified an inhibitor of the anti-apoptotic Bcl-2 family protein Mcl-1 that induced enhanced apoptotic capability combined with Verteporfin-PDT compared to treatment with either modality alone. In mice, this combination induced a stronger tumor growth inhibition than either modality alone, showing the feasibility of the approach.

### 9.2. Specific Targeting of the Vasculature

Another study used EGFP-EGF1 conjugated nanoparticles that encapsulated the PS hematoporphyrin mono-methylether (HMME) to target the vasculature for PDT [284]. The EGFP-EGF1 ensured preferential uptake by tissue factor-overexpressing cells which are aberrantly expressed on angiogenic vessels [285], enabling accumulation in the tumor vasculature versus non-conjugated particles in vivo as well as ex vivo. This tissue factor-targeting particle was tested in a follow-up study for the treatment of lymphoma in murine models [286], showing tumor growth inhibition in tumor-bearing mice. Another approach employed VTP with the PS IR700 dye by conjugation to a platelet-derived growth factor receptor b (PDGFR-b)-specific affibody [287]. PDGFR-b is abundantly expressed by the pericytes surrounding the tumor vasculature and is, therefore, a potential for vasculature targeting. The affibody-IR700 conjugate displayed a high affinity for PDGFRb and was shown to induce cell death in PDGFRb^+^ pericytes but not in PDGFRb- tumor cells. In mice, the conjugate was shown to distribute efficiently to tumors and to a slightly lesser extent to the liver and kidneys after intravenous injection. Ex vivo analysis showed co-localization of the conjugate with PDGFRb, confirming its targeting capacity. PDT using the PDGFRb-IR700 conjugate facilitated a tumor growth reduction in tumor-bearing mice, showing the feasibility of the approach. In a different study, the hypoxia-responsive prodrug TPZ was loaded into micelle aggregates consisting of the PS TPC (5-(4-carboxyphenyl)-10, 15, 20-tris (3-Hydroxyphenyl) chlorin) and a novel angiogenic-vessel targeting (AVT) cyclopeptide [288]. The rationale for this approach is to target PDT-induced angiogenesis and convert TPZ into its active form in the hypoxic tumor environment. In mice, the construct was shown to accumulate in tumors to a higher degree than particles lacking the AVT cyclopeptide. PDT with the construct induced a significant tumor growth inhibition compared to all control groups in mice bearing two tumors on opposite flanks.

### 9.3. Using VTP to Enhance Combination Treatments

In another study, VTP using the PS WST-11 was found to enhance the tumor retention of radioisotope ^90^Y-conjugated to DOTA-AR, a bombesin-antagonist peptide [289]. Bombesin is known to bind the gastrin-releasing peptide receptor (GRPr), which is overexpressed in multiple human cancers, including prostate cancer. Radiolabeled DOTA-AR was previously found to bind to GRPr and was specifically incorporated into PC-3 human prostate xenografts [290]. However, approximately two-thirds of injected radiolabeled DOTA-AR were washed out within the first 24 h. In order to improve this, VTP with WST-11 was applied before administration, improving the retention of ^90^Y-DOTA-AR in PC-3 tumors. Ex vivo analysis showed increased TUNEL and reduced CD31 staining, indicative of successful VTP. The combination of VTP and ^90^Y-DOTA-AR showed enhanced therapeutic efficacy in PC-3 tumor-bearing mice versus either treatment alone, indicating a synergistic effect. In another study, VTP using WST-11 has been combined with fractionated radiotherapy to improve treatment outcomes in prostate cancer models [291]. They show that VTP induces tumor growth inhibition with improved survival in tumor-bearing mice and that the combination of PDT with fractionated radiotherapy further enhances survival, providing a proof-of-concept for future studies. Recently, PDT with Radachlorin, a Ce6-based photosensitizer, was shown to enhance the accumulation of circulating NPs in murine models [114]. In mice, Radachlorin-PDT was shown to completely disrupt the vasculature and strongly inhibit tumor growth, resulting in the accumulation and retention of PEGylated poly-lactic-co-glycolic acid-based NPs in the tumor. Analysis of the tumor area revealed that the NPs distributed homogenously throughout the tumor area and that the majority of NPs are associated with immune cells of myeloid origin, among which are phagocytic cells. The results show the potential of combining VTP with NP-based therapeutics that benefit from targeting tumor-associated myeloid cells.

Together, these data show the potential of approaches that focus on increasing or exploiting the vasculature disruption induced by PDT. In addition, they underline the need for additional research to investigate more opportunities for improvements of VTP and combinations that further exploit the effects induced by the treatment.

## 10. Partial Destruction of the Tumor

An arduous challenge of PDT is a partial, incomplete destruction of the tumors that results in the survival of tumor cells after treatment, followed by rapid regrowth and tumor progression. One method to overcome this is by addressing the other challenges and opportunities of PDT that are described in this review or by improving the photosensitizer and protocol used for treatment. Another strategy to improve the destruction of the tumor is by combining PDT with other cytotoxic modalities, such as chemotherapy, to further reduce the number of viable cancer cells after treatment. As with the other efforts, there are many in vitro indications of the feasibility of this approach that precede testing in a preclinical setting [292,293,294,295,296,297,298,299,300].

### 10.1. Combinations with Chemotherapeutic Agents

In one study, the PS vinyl-substituted tetraphenylethylene (TPEPY) was integrated with the chemotherapeutic agent Mitomycin C (MMC), thereby quenching its activity as a PS and simultaneously inhibiting the cytotoxicity of MMC [301]. This prodrug construct was converted into its active form by glutathione, which is present inside cells, enabling its cytotoxic potential. In vitro, the conjugate caused a slight reduction in cancer cell viability in the absence of light and a strong reduction in viability in the presence of light, which could have been prevented by pretreatment of the cells with glutathione (GSH)-inhibitor BSO or ROS-scavenger vitamin C. After this, the conjugate was PEGylated to obtain NPs that were injected intratumorally in tumor-bearing mice. Following this, increasing intensity of TPEPY fluorescence was observed, showing the activation of the conjugate in vivo. PDT with the conjugate induced a statistically significant tumor growth inhibition compared to all relevant control, indicating the feasibility of this approach. Recently, the PS IR-780 dye and mitochondrial-acting anti-cancer drug lonidamine (LON) were co-encapsulated into cationic liposomes that localize to mitochondria after cellular uptake [302]. Illumination of these liposomes was shown to efficiently generate singlet oxygen and to raise the local temperature, causing LON to be released. The liposomes were shown to accumulate in murine tumors to a higher degree than in the liver, lung, and kidneys. PDT in tumor-bearing animals with the liposomes that contained IR780, with or without lonidamine, induced complete and lasting tumor regressions. These data show the antitumor potential of liposome-encapsulated IR780 dye but could not display the efficacy of LON in this setting, showing the need for additional research. Several approaches utilized nanosized carriers that contained a PS and a chemotherapeutic agent, e.g., pyropheophorbide and SN38, the active metabolite of topoisomerase inhibitor Irinotecan [303], paclitaxel and sinoporphyrin sodium [304], RGD functionalized UCNPs with pyropheophorbide a methyl ester and doxorubicin [305], that achieved similar results in murine models. They all show accumulation in tumors and tumor growth inhibition following PDT, further strengthening the potential of combining PDT with chemotherapy.

Another study investigated a highly complex particle consisting of magnetic mesoporous silica nanoparticles (M-MSNs) with the PS Ce6 and doxorubicin adsorbed onto its surface [306]. Moreover, alginate/chitosan polyelectrolyte multilayers were assembled around the surface to create pH-responsive particles, and shRNA for P-glycoprotein was additionally adsorbed onto this to alleviate multidrug resistance. The particles released their content more readily at decreasing pH and induced cytotoxicity to cancer cells In vitro. In tumor-bearing mice, PDT induced a significant tumor growth inhibition compared to controls. However, the benefit of the particles versus the particles without the pH-responsive layer and shRNA’s was minor, and the direct role of the shRNA for P-glycoprotein, as well as the pH-responsive layer, was not assessed properly, complicating an evaluation of their role in this setting. Although the strategy is interesting, additional research is required to determine whether this particle has a clear benefit over more simple approaches that are similarly effective. Of note, one study combined cisplatin chemotherapy with 5-ethylamino-9-diethyl-aminobenzo [a] phenothiazinium chloride (EtNBS)-mediated PDT [307]. Tumor-bearing mice with very large tumors (~800 mm^3^) were treated with EtNBS-PDT and induced a very strong tumor regression versus either modality alone, indicating the potential of this combination of large tumors, which are frequently resistant to therapy.

Several reports have utilized MOFs [308,309] as carriers for sensitizers and other antitumor agents, including antineoplastic agents [310]. One study created a MOF containing the chemotherapeutic agent pemetrexed and the PS 5-ALA for chemotherapy and photodynamic therapy [311]. The MOF reduced the viability of several cancer cell lines in the absence of light and without a chemotherapeutic agent, indicating toxicity of the particle itself. In HeLa tumor-bearing nude mice, PDT with the MOF induced a tumor growth inhibition, but the long-term therapeutic outcome was not assessed.

### 10.2. Combinations with Other Antineoplastic Agents

In a different study, the antitumor agent and heat-shock protein (HSP)90 inhibitor tanespimycin, in addition to the PS IR-820, were encapsulated into temperature-sensitive liposomes [312]. Illumination of the liposomes resulted in increased temperatures by IR-820-mediated PTT that triggered the release of tanespimycin. The liposomes induced light-dependent cytotoxicity in cancer cells and increased HSP90 expression. Intravenous administration of the liposome in tumor-bearing mice resulted in accumulation in the tumor in addition to the liver, lung, and kidneys. In mice, PDT treated after intratumoral administration of the liposomes induced a significant tumor growth inhibition versus all relevant controls. Unfortunately, the biological activity and antitumor effects of tanespimycin were not further investigated, complicating an accurate evaluation of its function in this setting.

When taken together, these studies show the feasibility and potential of combining PDT with other cytotoxic modalities to enhance the destruction of the tumor area. Many of the studies measured the antitumor efficacy only up to early timepoints after treatment, complicating an evaluation of the effects on survival and showing the need for additional research.

## 11. Insufficient PDT-Induced Antitumor Immune Responses Followed by Tumor Progression

Several successful efforts to enhance the therapeutic outcome after PDT utilizes the antitumor immune responses initiated by the treatment. This approach benefits from all aspects of PDT, which therein functions as a tumor-debulking and/or vasculature-disrupting modality that can induce acute inflammation in the tumor microenvironment that has been shown to result in tumor-specific responses [131,164,165,222]. Immunotherapy, in turn, is aided by the tumor-destructive capacities of PDT, which inhibits the tumor growth, disrupts the dense mass often observed in solid tumors, and can convert the immunosuppressive environment into a more inflammatory state [131,164,181,182,183,184,313,314]. PDT-induced antitumor immune responses have been shown to be essential for complete tumor clearance and progression-free survival in murine models, mostly by CD8^+^ T cell depletion studies that resulted in abrogation of the antitumor effect [131,154,156,165,168]. However, many tumors remain resistant to PDT, resulting in tumor outgrowth in spite of immune responses after treatment. Although it has been shown that some cancers display low endogenous levels of the DAMP calreticulin (CRT), thereby reducing phagocytic clearance and failing to induce immune responses [315], the exact mechanisms underlying the evasive mechanisms of tumors in the context of PDT are mostly unknown and may vary between tumor models. Several efforts have been undertaken to improve the efficacy of PDT-induced antitumor immune responses. These efforts can be divided into strategies that utilize PDT to generate or enhance tumor vaccines, strategies that combine PDT with various different forms of immunotherapy, and strategies that combine PDT with immune checkpoint inhibition.

### 11.1. PDT-Generated or Enhanced Tumor Vaccines

Several studies utilize PDT to induce ICD in cancer cells and enable access to previously inaccessible (neo)epitopes that initiates the maturation of dendritic cells (DCs). Such DCs, often called PDT-DCs, function as antitumor vaccines and are either generated in vitro by co-incubation of PDT-treated tumor cells with DCs or in situ after treatment of the tumor in vivo [25,31,157,158,158,159,316,316,317,318]. In one study, PDT-DCs were generated and used to treat glioma-bearing mice [319]. In vitro, PDT treatment of glioma cells was shown to induce surface exposure of the DAMPs CRT, HSP70, and HSP90 in addition to an increase in extracellular DAMPs adenosine triphosphate (ATP) and high mobility group box 1 (HMGB1). PDT-DCs were then generated by treating glioma cells with Hypericin (Hyp)-PDT, after which the cells were co-incubated with BMDCs. In a prophylactic setting, high survival rates were observed versus no surviving animals for control or mice treated with freeze/thaw (F/T, as a model for necrosis) incubated DCs. Neutralization of either HGMB1, CRT, extracellular ATP, or treatment with antioxidants all reduced the efficacy of the PDT-DC vaccine, showing the importance of DAMPs as well as PDT in this setting. Moreover, administration of PDT-treated tumor cells in absence of DCs significantly reduced mouse survival rates, indicating enhanced efficacy for PDT-DCs over injection of PDT-treated tumor cells. Furthermore, Rag1^−/−^ mice did not respond to treatment with PDT-DCs, underlining the importance of the adaptive immune system. Brain-infiltrating immune cells after PDT-DC treatment showed increases in total T cells, CD4^+^ T cells, CD8^+^ T cells and Th17 cells as well as reduced amounts of regulatory T cells (Tregs) compared to control mice. In a therapeutic setting, PDT-DC treatment was combined with temozolomide (TMZ) chemotherapy and induced a strong increase in survival versus either modality alone. As expected, the TMZ treatment was found to reduce the absolute numbers of intra-brain mononuclear cells and CD8+ T cells. However, the PDT-DC vaccine treatment reversed this effect for mononuclear cells, but not for CD8^+^ T cells. Furthermore, the PDT-DC treatment reduced the amount of brain Tregs compared to control and treatment with TMZ alone. These results show the potential of PDT-generated DC vaccines in the treatment of glioma-bearing mice and underline the importance of DAMP generation and the presence of a functional adaptive immune system for the efficacy of such treatments.

In another study, Hyp-PDT was shown to enhance surface exposure of CRT, HSP70, and HSP90 and reduce the levels of the “don’t eat me” signal CD47 [155]. Moreover, Hyp PDT induced phagocytosis of cancer cells by DCs and induced upregulation of maturation markers CD80, CD86 and CD40 to a larger extent than DCs co-cultured with F/T treated cancer cells. In animal models, the PDT-DCs were found to be potent inducers of IFN-γ-secreting CD8^+^ T cells from autologous T cells and initiated a reduction in the total amount of CD4^+^ CD25^+^ Foxp3 cells. Furthermore, the PDT-DCs was shown to inhibit tumor growth in a prophylactic setting, strongly enhancing survival versus F/T-treated LLCs and other relevant controls. CTLs obtained from the immunized mice were shown to efficiently induce cancer cell death ex vivo for PDT-DC mice, significantly enhanced compared to mice vaccinated with PDT-treated cancer cells lacking DCs, indicating the existence of tumor-specific T cells after treatment with PDT-DC. In another study, the effect of light fluence on the functional maturation of DCs was investigated [158]. To this end, cancer cells were treated with 5-ALA-PDT at different fluences ranging from 0.125–2 J/cm^2^. A fluence of 0.5 J/cm^2^ was shown to induce the largest proportion of early apoptotic cells of all fluences tested, which subsequently induced the highest IFN-γ production in BMDCs after co-incubation. Furthermore, this early apoptosis-inducing PDT regimen was shown to induce morphological hallmarks of DC maturation and displayed strong upregulation of maturation markers MHC-II, CD80, and CD86. In addition, mice treated with the PDT-DCs generated by the early-apoptosis PDT regimen were protected from tumor challenge, whereas animals vaccinated with F/T-DCs were not. These results indicate that 5-ALA PDT at a regimen that induces a high proportion of apoptotic cells can induce strong DC maturation that can prevent tumor outgrowth in a prophylactic setting. Together, these results strongly indicate that PDT-induced oxidative stress can exert potent immune stimulation and that vaccination with PDT-treated tumor cells, administered directly or after co-incubation with DCs, can reduce tumor growth and enhance survival.

Taking a different approach, Kleinovink et al. performed a study that combined PDT with tumor-specific vaccination against TC-1 and RMA tumors [165]. Radachlorin-PDT induced a tumor growth delay in TC-1 tumors without inducing complete responses. Serum analysis showed a significant increase in HMGB-1 serum levels in PDT-treated mice compared to the control. The tumor-draining lymph nodes displayed an increase in total numbers of CD8^+^ T cells, tumor (TC-1)-specific CD8^+^ T cells, and CD11c^+^ cells versus non-tumor draining lymph nodes in PDT-treated and versus dLNs and ndLNs in control mice. In a therapeutic setting, a combination of PDT with vaccination significantly improved the survival of mice compared to either treatment alone, showing the potential of the approach. Moreover, all cured animals rejected a secondary tumor challenge, indicating the existence of immunological memory. In a distant tumor model, the combination also enhanced the survival of tumor-bearing mice compared to either treatment alone.

Together, the data show the efficacy of combining PDT with cancer vaccinations for treating primary as well as metastatic tumors.

### 11.2. Combination with Immunostimulatory Agents

Several studies have effectively combined PDT with immunotherapeutic agents that improve antitumor efficacy. In one study, exogenously administered CRT was used to boost PDT-generated immune responses [320]. The antitumor efficacy of Ce6-treated cancer cells injected into tumor-bearing mice was enhanced by pre-incubation of the cancer cells with recombinant CRT or cell surface CRT-inducing agent mitoxantrone prior to injection. In addition, the tumor response to mTHPC-PDT was shown to be significantly enhanced by administering CRT as an adjuvant. The same effect was not observed in NOD-SCID mice, underlining the role of the immune system in this process. Another approach attempted to target Tregs for death with PDT by using anti-CD25 antibodies conjugated to Ce6 [321]. The antibody was shown to bind to CD25^+^ CD4^+^ T cells after intravenous administration in murine models. PDT effectively depleted CD25^+^ CD4^+^ T cells and led to an increase in infiltrating CD8^+^ T cells, but not CD4^+^ T cells, compared to isotype-Ce6, anti-CD25 alone, and untreated. Furthermore, the amount of intratumoral IFN-y-producing CD8^+^ T cells and IFN-y^+^CD107a^+^CD8^+^ cytotoxic T cells were increased by the regimen. PDT also induced a significant tumor growth inhibition in mice compared to isotype-Ce6, anti-CD25 alone, and untreated. Another approach consisted of mitochondria-directing particles that contained the PS IR-820 and the toll-like receptor (TLR)-ligand CpG for PDT and immunotherapy [322]. This particle displayed mitochondria enrichment and was able to induce strong cancer cell death, while the CpG in the particles was shown to retain biological activity. In mice, the particle accumulated in the tumor and PDT induced significant growth inhibition compared to the particle without CpG, showing the importance of CpG in this setting. Recently, the efficacy of Radachlorin-PDT combined with NPs containing two TLR ligands and a leukocyte-attracting agent was investigated [131]. The combination induced strong antitumor responses in several murine tumors, significantly enhancing survival and inducing an abscopal effect in distant tumors. The observed effects were shown to depend on the presence of CD8^+^ T cells, as depletion completely abrogated the antitumor efficacy. Moreover, the combination was reported to convert the immunosuppressive tumor microenvironment from cold (immunosuppressed) to hot (pro-inflammatory). Finally, the treatment was shown to function as an in-situ vaccination modality that induced tumor-specific, oncoviral- or neoepitope-directed, CD8+ T cells against the respective tumors.

In another study, a core consisting of the IDO inhibitor 1-methyltryptophan (1MT) was coupled to the PS PpIX through a peptide sequence that is cleaved by caspase-3 for (PDT-) inducible release of 1MT [323]. PDT with the construct induced an enhanced tumor growth inhibition compared to treatment with either PpIX-PDT or 1MT treatment alone. Furthermore, the regimen reduced the number of metastatic tumor nodules in the lung treatment, suggesting an abscopal effect. Analysis of immune cell populations in blood and spleen revealed reduced percentages of CD4^+^ T cells and increased CD8/CD4 ratio, indicating 1MT activity in vivo. Finally, TNF-a, IFN-y, and IL-17 were increased, while IL-10 was reduced after treatment in both primary and metastatic lung tumors. Similarly, a chlorin-based particle was created that also contained an IDO inhibitor, displaying strong cytotoxicity in several cancer cell lines [324]. In mice, PDT with the particles induced a significant tumor growth inhibition in treated and distant tumors, whereas the PDT with the particles without the IDO inhibitor induced a tumor growth inhibition on treated but not untreated tumors. Percentages of CD45^+^ cells and CD4^+^ T cells were increased in both treated and untreated tumors after PDT; CD8^+^ T cells were only increased in the untreated tumors, while B cells as well as neutrophils were only increased in the treated tumors. These results show the potential of combining PDT with IDO inhibitors and suggest a strong involvement of the immune system in therapeutic efficacy.

These papers show that PDT combines well with different forms of immunotherapy, underlining the ability of immunotherapy to complement and enhance the antitumor efficacy of PDT.

### 11.3. Combination with Immune Checkpoint Blockade Antibodies

Several studies have combined PDT with immune checkpoint blockade antibodies to enhance the antitumor efficacy of PDT. To this end, UCNPs containing PS Ce6 and TLR-7 agonist imiquimod (R837) were employed and combined with anti-CTLA-4 antibodies [180]. In vitro, PDT with the particles induced cancer cell death, and incubation of PDT-treated cancer cells with BMDC induced upregulation of maturation markers CD80 and CD86. In mice, PDT with the particles on tumors induced DC maturation in the tumor-draining lymph nodes and elevated blood levels of IL-12p40, IFN-y, and TNF-a 3 days after treatment. Moreover, PDT combined with CTLA-4 treated on mice bearing two tumors on opposite flanks induced complete and lasting responses in both treated and distant (untreated) tumors, in contrast to all relevant controls. The addition of anti-CTLA-4 antibody treatment to PDT treatment with the particle-induced increased amounts of CD8^+^ T cells and reduced the numbers of Tregs in tumor infiltrates. In addition, PDT combined with anti-CTLA-4 antibody treatment led to elevated IFN-y levels in serum versus PDT treatment alone. As a control for immunological memory, cured mice were rechallenged with C26 tumors, after which the majority of mice were protected from tumor challenge. Another approach consisted of VTP with WST-11 combined with anti-PD-1/PD-L1 antibodies [325]. In tumor-bearing mice, only the full combination provided a significant tumor growth delay in addition to increased progression-free survival. Furthermore, the combination led to an increase in the CD8^+^:Treg and conventional T cells (Tconv):Treg ratios and was shown to reduce the number of metastatic lesions in the lung compared to either modality alone. In distant tumors, infiltrating lymphocyte populations were analyzed, but no significant differences were found. Moreover, CD8:Treg and Tconv:Treg ratios appeared to be lower in the distant tumors for the combination compared to VTP alone, and no differences in proliferating T cells were shown. Lastly, human xenografts were shown to upregulate expression of PD-L1 after VTP with WST-11, suggesting a rationale for initiating human trials investigating the combination of VTP and anti-PD-L1 antibody treatment.

Several other studies successfully combined PDT with anti-PD-L1 antibodies [326,327]. One such study combined the PS Fe-5,10,15,20-tetra(p-benzoato)porphyrin (TBP) with anti-PD-L1 antibody treatment [328]. Fe-TBP produced singlet oxygen under both normoxic and hypoxic conditions due to its ability to convert H_2_O_2_ to O_2_, which can subsequently be used to yield singlet oxygen. In mice, PDT with Fe-TBP induced complete regressions in tumor-bearing animals. These regressions were shown to be CD4^+^ T cell, CD8^+^ T cell, and B cell-dependent, as depletion of these cells significantly diminished the antitumor effects. Furthermore, Fe-TBP PDT on primary tumors combined with anti-PD-L1 antibodies also strongly inhibited the growth of distant tumors versus both treatments alone. Cured mice were protected from tumor rechallenge, indicating the existence of immunological memory. Lastly, PDT and anti-PD-L1 antibody treatment induced increased amounts of total CD45^+^ cells in primary tumors as well as increased amounts of CD4^+^ and CD8^+^ T cells in primary and distant tumors versus untreated animals, further indicating the involvement of the immune system in the treatment response. Similarly, a different study investigated pH-responsive PEGylated NPs with a mitochondria-directing agent that encapsulate catalase enzymes to alleviate hypoxia by conversion of H_2_O_2_ to O_2_ were loaded with Ce6 for PDT [329]. These particles displayed preferential localization to mitochondria and induced efficient light-dependent toxicity to cancer cells in hypoxic areas compared to particles lacking catalase. Accumulation of the particle was observed mostly in the liver, followed by the tumor, but PDT still induced a significant tumor growth inhibition compared to the relevant controls. Combination of PDT and anti-PD-L1 antibody treatment induced significant tumor growth inhibition on both treated and distant tumors, whereas PDT with the particle in the absence of anti-PD-L1 antibody treatment only induced significant tumor growth inhibition on primary (treated) tumors. Furthermore, this combined regimen increased the percentages of CD8^+^ T cells in the tumor and IFN-y levels in sera after treatment, indicating the involvement of the immune system. A different study investigated core-shell NPs encapsulating chemotherapeutic agent oxaliplatin and the PS pyrolipid combined with anti-PD-L1 antibodies [179]. In vitro, PDT with the NPs induced ICD through increased exposure of CRT in cancer cells. In tumor-bearing mice, PDT induced tumor growth inhibition in two different models and increased the serum levels of IFN-y, IL-6, and TNF-a after treatment. Furthermore, PDT performed on primary tumors combined with anti-PD-L1 antibodies induced strong tumor growth inhibition in both primary and distant tumors compared to two different models. In another study, zinc porphyrin silica NPs loaded with R837 and combined with anti-PD-L1 antibodies [330]. The R837 was released at low pH and promoted dendritic cell maturation, inducing the conversion of the tumor microenvironment into an inflammatory state. PDT with the NPs combined with anti-PD-L1 induced strong antitumor effects and an abscopal effect and was shown to increase the CD8^+^/CD4^+^ ratio as well as the percentage of CTL in both primary and distant tumors.

Together, these studies show that PDT combined with checkpoint blockade antibodies is a highly effective treatment option with strong antitumor efficacy on both primary and models of metastatic tumors.

The current challenges and recent attempts for improvements are summarized in Table 2.

## 12. Recent Advances in Clinical Photodynamic Therapy

The use of PDT as a standalone treatment or in combination, in trials and in the clinic, has been summarized previously [41,42,331,332]. Recent clinical trials include combining interstitial PDT using porfimer sodium with the standard of care chemotherapy or immunotherapy in patients with locally advanced or recurrent head and neck cancer (NCT03727061). This trial is currently recruiting and could provide valuable insights into the efficacy of interstitial PDT in patients when combined with chemotherapy or immunotherapy, possibly allowing a comparison of these combinations. One other trial is investigating the efficacy of 5-ALA PDT combined with the anti-PD1 antibody Nivolumab in patients with malignant pleural mesothelioma (NCT04400539). This pilot trial is not yet recruiting but it could be pivotal in determining the efficacy of PDT combined with immune checkpoint inhibition in humans, which has shown great promise in preclinical models. A different trial is investigating PDT using porfimer in an intraoperative setting sodium and immune checkpoint inhibition in patients with non-small cell lung cancer that display pleural disease (NCT04836429). This trial is currently being recruited and could improve our understanding of PDT-induced immune stimulation in an intraoperative setting. Finally, a trial dedicated to understanding the immune response following 5-ALA PDT in patients with basal cell carcinoma has been initiated (NCT05020912). This trial is currently recruiting and determines several immunological response parameters after treatment, possibly providing novel insights related to the immune response induced by PDT in humans.

## 13. Concluding Remarks

The field of PDT has been developing at a steady pace, enhancing the treatment efficacy by addressing one or several of the limitations of PDT. Many improvements have been made related to the biodistribution of PS that enhances their accumulation in the tumor. This increases the number of PS in the tumor and, therefore, theoretically, also the ability of these PS to induce damage to the tumor area. Improvements related to the penetration depth of light used for treatment have also been made, increasing the area where the photodynamic effect can occur in larger tumors. Moreover, interesting PS have been developed that partially alleviate or circumvent the hypoxic state present in the tumors, which may increase the damage to cells in the tumor area. These efforts all enhance the direct tumor-killing capacity of PDT, reducing the number of viable cancer cells after treatment compared to previously applied PDT modalities. In addition, a combination of PDT with other antineoplastic agents can further enhance tumor growth inhibition and improve survival after treatment. Many of these combinations were shown to be effective against primary tumors but were not shown to induce an abscopal effect. In this regard, combinations of PDT and immunotherapy were highly effective, inhibiting the growth of both primary and distant metastatic tumors in a preclinical setting. Although some trials that investigate PDT with immunotherapy have been initiated, the combination has not been thoroughly investigated in humans. Future trials will have to determine whether the efficacy observed in a preclinical setting reflects the treatment outcome in the clinic.

In the coming years, additional investigation into the mechanism behind the immunostimulatory capacities of PDT is instrumental to further enhancing its efficacy and providing a more solid basis to translate the combination of PDT and immunotherapy to the clinic. Improving the efficacy of PDT in that regard may involve enhancing its ability to induce ICD to initiate an increased acute inflammation in the tumor and provide an optimal environment for the immune system to clear the remaining tumor cells. Moreover, additional knowledge of the optimal regimen of PDT and immunotherapy will increase the chances that the best combination is tested in a clinical setting. These investigations combined could further the translation of PDT into the clinic. Alternatively, as the fields of PDT and immunotherapy keep evolving, exciting novel treatments may emerge that result in enhanced treatment outcomes in patients.

## Figures and Tables

**Figure 1 pharmaceutics-15-00330-f001:**
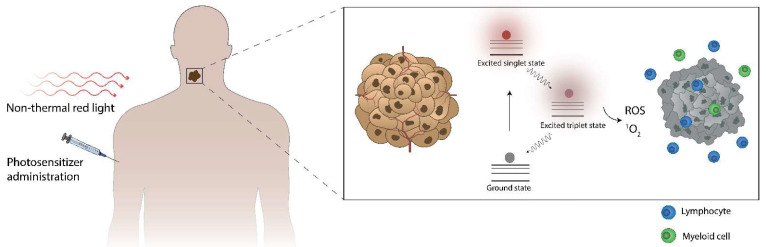
A short summary of photodynamic therapy.

**Figure 2 pharmaceutics-15-00330-f002:**
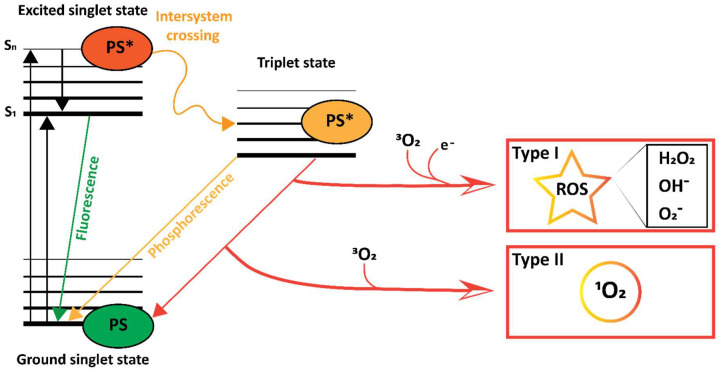
Photodynamic effect.

**Figure 3 pharmaceutics-15-00330-f003:**
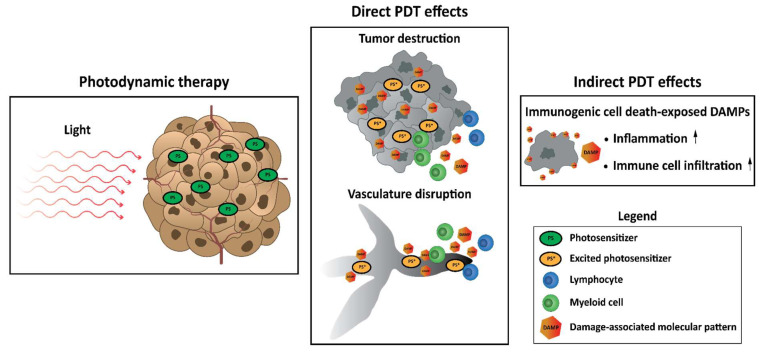
Consequences of photodynamic therapy.

**Figure 4 pharmaceutics-15-00330-f004:**
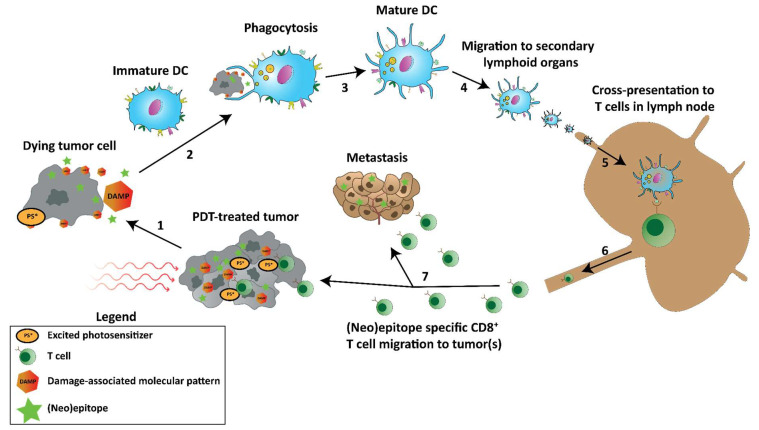
Tumor-specific immune responses induced by PDT.

**Figure 5 pharmaceutics-15-00330-f005:**
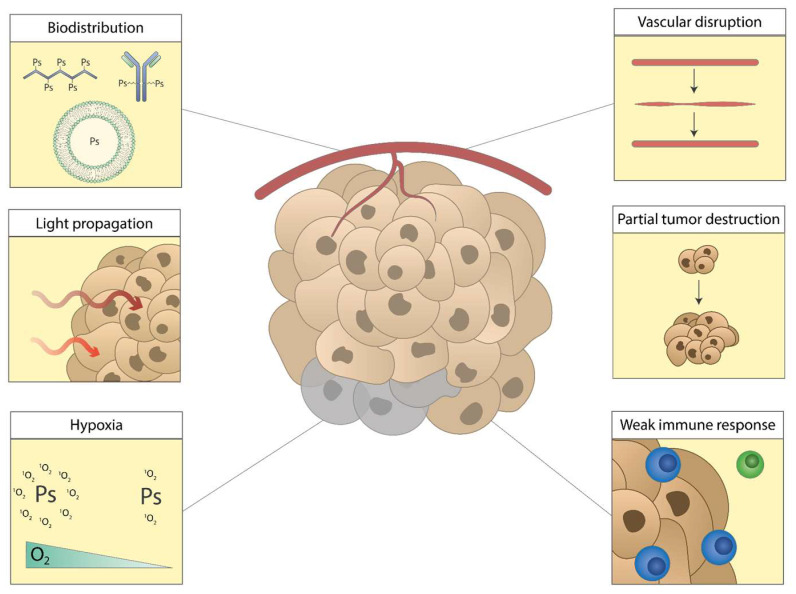
Current challenges of photodynamic therapy.

**Table 1 pharmaceutics-15-00330-t001:** Overview of notable clinically used photosensitizers.

Sensitizer (Brand Name)	Approval	Indication	Wavelength (nm)
5-aminolevulinic acid/5-ALA (Ameluz) (Levulan)	Worldwide	Mild to moderate actinic keratosis	635
Bremachlorin (Radachlorin)	Russia, Belarus	Non-small cell lung cancer, bladder, cutaneous lesions	662
Hexaminolevulinate hydrochloride (Cysview)	Europe, USA, Canada	Bladder cancer detection	360–450
Methyl aminolevulinate (Metvix) (Metvixia)	Worldwide	Non-hyperkeratotic actinic keratosis and basal cell carcinoma	570–670
Porfimer sodium (Photofrin)	Worldwide	Esophageal cancer, Barrett’s Esophagus, non-small cell lung cancer	630
Redaporfin (LUZ11)	Orphan status in Europe	Biliary tract cancer	749
Synthetic hypericin (SGX301)	Orphan status in Europe, conditional FDA	Early-stage cutaneous T-cell lymphoma	570–650
Talaporfin sodium/NPe6 (Laserphyrin)	Japan	Lung cancer	664
Temoporfin/mTHPC (Foscan)	Europe	Advanced Head and neck cancer	652
Verteporfin (Visudyne)	Worldwide	Age-related macular degeneration	690
WST-11/padeliporfin (TOOKAD)	Europe	Prostatic Neoplasms	753

**Table 2 pharmaceutics-15-00330-t002:** Overview of the advantages and disadvantages for the current opportunities of PDT.

Challenge	Strategy to Overcome	Advantages	Disadvantages
Sensitizer biodistribution	Nanoparticle encapsulation	Control over pharmacokinetics, possibility of adding targeting moieties.	The enhanced-permeability and retention effect is much less pronounced in humans.
	Antibody conjugation	Specific targeting of tumor epitopes, adoption of antibody biodistribution.	Lack of truly specific tumor targets.
	Peptide association	Targeting of tumor present ligands, adoption of peptide association.	Lack of truly specific ligands in the tumor.
	EV incorporation	Enhanced biodistribution of the PS, enhanced antitumor efficacy depending on the EV origin.	Large scale production is challenging. Restricted to the use of cell lines.
	Immune cells	Tunable distribution based on cell type and immunological state. Possibility to simultaneously use immune cells for therapy.	Restricted to distribution and functionality of immune cells in use. Distribution of PS to tumor cells required following death of carrier cells.
Light propagation	NIR absorbing sensitizers	Increased penetration depth of light used for PDT.	High fluence required due to a reduced energy of therapeutic NIR light.
	Upconversion Nanoparticles	Increased penetration depth of light used for PDT. Possibility to co-encapsulate additional therapeutic agents.	High fluence required due to a reduced energy of therapeutic NIR light.
Hypoxia	O_2_-generating strategies	Possibility to increase ROS quantum yields after PDT.	Requires the use of carrier systems for O_2_-generating agents.
	Hypoxia-responsive prodrugs	Drug selectivity to hypoxic areas in the body, such as the tumor.	Restricted to certain prodrugs that are hypoxia-responsive.
	O_2_ tumor diffusion	Increased availability of O_2_ for PDT throughout the tumor.	Requires PS or PDT protocols that can be directed to the ECM.
Vascular disruption	Tumor vasculature disrupting agents	Enhanced tumor vasculature disruption.	Risk of adverse events of vasculature-disrupting agents.
	Tumor vasculature targeting	Enhanced tumor vasculature disruption.	Risk of adverse events due to vasculature-destruction.
	VTP to enhance combination treatments	Increased potential for synergy with additional agent due to vasculature disruption.	Efficacy depending on ability of VTP to sufficiently disrupt the vasculature.
Partial tumor destruction	Combination with chemotherapy	Increased antitumor efficacy.	Associated with a higher risk of adverse events.
	Combination with other neoplastic agents	Increased antitumor efficacy.	Depending on the agent used, but often associated with increased risk of adverse events.
Insufficient PDT-induced immune response	PDT-generated or enhanced tumor vaccines	Possibility to generate in situ vaccinations, or to enhance vaccination efficacy. Possibility to affect metastatic tumors.	Efficacy of the treatment is dependent on the ability of PDT to induce a pro-inflammatory environment in the tumor.
	Combination with immunostimulatory agents	Increased antitumor efficacy, possibility to generate an in situ vaccination. Possibility to affect metastatic tumors.	Certain immunostimulatory compounds require encapsulation to prevent adverse events.
	Combination with immune checkpoint inhibition	Increased antitumor efficacy. Possibility to affect metastatic tumors.	Increased risk of adverse events

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
