# Peer review of "Current Challenges and Opportunities of Photodynamic Therapy against Cancer"

_pharmaceutics, 2023, doi:10.3390/pharmaceutics15020330_

Round 1

Reviewer 1 Report

The article by Huis in’t Veld et al.  provides an extensive review of photodynamic therapy (PDT) strategies developed to date, highlighting the challenges and opportunities of such a technique.  The article is well-written and a comprehensive view of PDT is provided. I am able to recommend the publication of the article in Pharmaceutics upon completion of a few minor revisions:

1)    In Figure 1 the abbreviation PS is used, but the text where the abbreviation is described comes after the figure, so the authors are advised to change this.

2)    The authors should use italic font for in-vitroin-vivo thorough the manuscript.

3)    The authors are advised to summarize all the possible strategies (peptides, nanoparticles ecc.) in a table, with advantages and disadvantages highlighted. This would help the reader.

Author Response

1)    In Figure 1 the abbreviation PS is used, but the text where the abbreviation is described comes after the figure, so the authors are advised to change this.

 Thank you for pointing this out. We have clarified this in the updated manuscript.

2)    The authors should use italic font for in-vitroin-vivo thorough the manuscript.

Noted, we have changed this in the altered manuscript.

3)    The authors are advised to summarize all the possible strategies (peptides, nanoparticles ecc.) in a table, with advantages and disadvantages highlighted. This would help the reader.

We agree and have added this table in the updated manuscript.

Reviewer 2 Report

1.             The quality of fig 1 should be improved. Also, the author should describe the mechanism in detail.

2.             From the Photodynamic Therapy, how about the types of the photosensitizers (PS), the authors should describe and discuss

3.             What potential does further research hold? What is the ultimate goal in this field?

4.             Does the future of study lie in this area? Are there other more promising areas in the field which could be progressed?

5.             How will the field evolve in the future? In your perspective, what will the standard procedure have gained or lost from the current norm in five or ten years?

6.             There are some updated refs and reviews could be highlighted, such as

7.             Dalton Transactions, 2022, 51, 14817-14832; J. Mater. Chem. B., 2022, 10, 5105 – 5128; Colloid Surface B, 2022, 213, 112432; J. Colloid. Interf. Sci, 2022, 621, 180-194 and New J. Chem., 2021, 45, 20987–21000 and Inorganics, 10(2022) 202

8.             I suggest the authors added more examples and figures on this section (Partial destruction of the tumor), some PDT, SDT, PTT combination

Author Response

  1. The quality of fig 1 should be improved. Also, the author should describe the mechanism in detail.

Thank you for your suggestion. We have increased the quality updated the quality of the figure. Please note that this is a schematic summary of PDT and that the mechanism is explained in depth throughout the manuscript.

  1. From the Photodynamic Therapy, how about the types of the photosensitizers (PS), the authors should describe and discuss

We have added the types of sensitizers and elaborated on them in the updated manuscript. Please note that an in-depth assessment of the types of sensitizers is beyond the scope of the current review, which is relatively lengthy in its current state.

  1. What potential does further research hold? What is the ultimate goal in this field?

The ultimate goal is to cure cancer, preferably in a manner that is the least invasive/restrictive to patients.

  1. Does the future of study lie in this area? Are there other more promising areas in the field which could be progressed?

In the preclinical setting, PDT (using various PS from different generations) as a standalone treatment option has shown strong antitumor efficacy, with complete and lasting cures for several different PS that resulted in investigations in the clinic. However, PDT as a standalone treatment option subsequently has shown limited efficacy in the clinic (depending on the type of lesion) in spite of highly curative and promising results in the preclinical setting. We currently believe (based on our results as well as the recent developments in the field) that the most potential lies in combination with treatments that benefit from the mechanism of action of PDT, rather than relying on optimizing the tumor cell-killing potential of PDT alone. As the fields of PDT and other types of cancer treatments with which PDT combines well progress simultaneously, this strategy may benefit from improvements in all those fields. More specifically, we believe that there is great potential in combinations of PDT and immunotherapy due to the tumor-debulking and immune-stimulating capacities of PDT that can be further enhanced when using the right immunotherapy at a correctly calibrated time after PDT treatment. There are also many other promising areas in the field of cancer treatment that we do not fully address here (eg., TKIs, adoptive cell therapies, image-guided surgery etc.). However, describing all novel cancer treatments falls beyond the scope of this review.

  1. How will the field evolve in the future? In your perspective, what will the standard procedure have gained or lost from the current norm in five or ten years?

In the future perspectives, we have further updated a possible future for the field. However, we would like to steer clear of strong predictions of the future as that has proven to be only partially true (at best) in a rapidly evolving and expanding field of research.

  1. There are some updated refs and reviews could be highlighted, such as
  2. Dalton Transactions, 2022, 51, 14817-14832 J. Mater. Chem. B., 2022, 10, 5105 – 5128; Colloid Surface B, 2022, 213, 112432; J. Colloid. Interf. Sci, 2022, 621, 180-194 and New J. Chem., 2021, 45, 20987–21000 and Inorganics, 10(2022) 202

We have analyzed the suggested references and added those that are relevant to the manuscript.

  1. I suggest the authors added more examples and figures on this section (Partial destruction of the tumor), some PDT, SDT, PTT combination

The review contains literature that describes interesting uses of PTT in the context of PDT. Unfortunately, we were unable to find a suitable paper with the combination of PDT, PTT and SDT. We were able to find papers that use PDT and PTT. However, those often showed that PTT induced a form of heating in vitro, but lacked the proper controls to evaluate if PTT was truly synergizing with PDT in animal models (e.g., Adv. Sci. 2020, 7, 2001088, compared PDT + PTT to only one round of PDT or PTT instead of two rounds of PDT or PTT).We would also like to emphasize that the current review focuses on the challenges of PDT and successful combinations thereof, rather than on SDT and PTT.